

# The degree of freedom for signal assessment of measurement networks for joint chemical state and emission analysis

Xueran Wu[1,2,3], Hendrik Elbern[1,2], and Birgit Jacob[3]

[1]IEK-8, Forschungszentrum Jülich, Wilhelm-Johnen-Straße, 52428, Jülich, Germany.
[2]Rhenish Institute for Environmental Research at University of Cologne, Aachener Straße 209, 50931, Cologne, Germany.
[3]Mathematics Department, Unversity of Wuppertal, Gaußstraße 20, 42119, Wuppertal, Germany.

*Correspondence to:* Xueran Wu (x.wu@fz-juelich.de)

**Abstract.** The Degree of Freedom for Signal (DFS) is generalized and applied to estimate the potential observability of observation networks for augmented model state and parameter estimations. The control of predictive geophysical model systems by measurements is dependent on a sufficient observational basis. Control parameters may include prognostic state variables, mostly the initial values, and insufficiently known model parameters, to which the simulation is sensitive. As for chemistry-transport models, emission rates are at least as important as initial values for model evolution control. Extending the optimisation parameter set must be met by observation networks, which allows for controlling the entire optimisation task. In this paper, we introduce a DFS based approach with respect to address both, emission rates and initial value observability. By applying a Kalman smoother, a quantitative assessment method on the efficiency of observation configurations is developed based on the singular value decomposition. For practical reasons an ensemble based version is derived for covariance modelling. The observability analysis tool can be generalized to additional model parameters.

## 1 Introduction

Air quality and climate change are influenced by the fluxes of green house gases, reactive gas emissions and aerosols in the atmosphere. The temporal evolution of chemistry in the atmosphere is usually modelled by atmospheric chemistry transport models. Numerical chemistry transport models include solvers of the reaction-diffusion-transport equation as the central part, hence presenting the solution of an initial-boundary value problem. The insufficiency of well-known initial values, sources and sinks is therefore a serious problem for the quality of simulation, which can be addressed by data assimilation and inverse modelling. The question to what extend the forecast skill of a prognostic model by which erroneous parameter can be only answered if the observation network is sufficiently sensitive to their mis-specifications. This poses the observability problem. However, in many cases, better estimates of both the initial state and emission rates are not always neccessarily guaranteed by appropriate observational network configurations. Then it may hamper the optimization by erroneous modifications of initial state and emission rates, which will result in degraded simulations beyond the time interval with available observations. In fact, the ability to evaluate the suitability of an observational network to control chemical states and emission rates is a key qualification, which needs to be addressed when designing elaborate and costly field campaigns or permanently operational observation networks.



In practice data assimilation problems are typically solved in circumstances, where the number of observations are markedly lower than the model degree of freedom Daley (1991), as implied by its numerical set-up. Consequently, when aiming to improve the quality of analyses by the observation configuration, several aspects can be considered. These include (i) to optimise the observation network, subject to given constraints, (ii) to evaluate the value of individual or types of observations for the analyses, and (iii) to quantify the degree of which the analysis can be influenced by the observations, tat is the sensitivity.

The observation network optimisation problem (i) has been addressed traditionally by Observation System Simulation Experiments (OSSEs, see for example (Daley, 1991)). The advanced concept of targeted observations has been popularized with the FASTEX campaign (e.g Szunyogh et al. 1999; Langland et al. 1999). Theoretical studies are presented, for example by Bishop et al. (1999); Berliner et al. (1998), or recently by Bellsky et al. 2014 for highly nonlinear dynamics anfd Wu et al. (2016) for the optimal locations of observations for time-varying system within a finite-time interval.

The benefit assessment (ii) of individual or types of measurements or remote sensing data within a given network seeks to identify the value ranking of information sources, accounting for analyses achieved. This problem has been investigated by Cardinali et al. (2004); Cardinali (2009), and a sequence of related papers , or Liu and Kalnay (2008), the latter without use of an adjoint model. A related approach was described by Baker and Daley (2000), who exploited sensitivities to observations to identify their spatial extensions of impact.

Finally, the need to quantify the information content provided by the observations (iii) can be satisfied by suitable and calculable measures, such as entropy reduction or degrees of freedom for signal. Fisher (2003) presented two practically useful methods to infer, in addition to the entropy reduction, the DFS achieved by assimilating many thousands of observations in the European Centre of Medium Range Weather Forecast (ECMWF) 4D-var system. Prior to this study, the concept of DFS has been applied to satellite retrieval problems, typically lower dimensional as compared to data assimilation (see for example Eyre 1990; Rodgers 2000; Rabier et al. 2002; Fourrié et al. 2003; Martynenko et al. 2010 ).

The DFS concept is able to identify the potential of measurement networks independent of actual data. This assures that the assessment of the impact of observations is independent from the difference between the analysed and the forecasted or background state. However, these studies focussed on the classical data assimilation problem with initial values or prognostic state variables being the only parameters of optimisation. Especially for chemistry transport or greenhouse gas models with high dependence on the emissions in the troposphere, the optimization of initial state is no longer the only issue. In order to get better analyses from combining the model with observations, efforts of joint optimization by adding the emission rates to concentrations have been made (Elbern et al. 2000, 2007; Bocquet 2012; Bocquet and Sakov 2013; Miyazaki et al. 2012; Tang et al. 2011, 2013; Winiarek et al. 2014). Yet the lack of ability to observe and estimate surface emission fluxes directly, with necessary spatial density, is a major roadblock, hampering the progress in predictive skills of climate and atmospheric chemistry models. So area flux measurements are only available by eddy covariance observations of green house gases by a sparse network of towers. Therefore the capacity to distinguish between the degree of freedom for signal of both emission rates and concentrations is crucial to assess the value of a measurement network. This assessment is however dependent not only on observation network and its deployment with respect to emission sources, but also the assimilation window lengths and the meteorological transport conditions.





A meanwhile classical task is greenhouse gas inversion, aiming at the estimation of carbon dioxide, methane, and nitrous oxide, from which a rich set of literature emerged. For example, in case of $CO_2$, Peter et al. (2005) devised an ensemble data assimilation approach, approximating the covariance matrix without need to use an adjoint model version. In a more general context, the emission rate optimisation problem may be considered as an analogue to identify model errors. The sensitivity of the model evolution with respect to the model errors and the observation network as its detector is the a key quantity to be analysed. Several methodologies have been formulated to account for model errors in both variational and ensemble data assimilation (e.g. Bellsky et al. 2014; Gillijns and De Moor 2007; Li et al. 2009; Smith et al. 2013; Tremolet 2007). With focus on the observability, Daescu (2004) presented a method, resting on only one additional adjoint model integration for measurement network optimization. Cioaca and Sandu (2014) introduced a general framework to optimize a set of parameters controlling the 4D-var data assimilation system, which includes means to identify erroneous data, observation accuracy and location. In a related paper Cioaca and Sandu (2014) quantified the observation impact in terms of reduction of uncertainties of shallow model state and other parameters. Navon (1997) outlined the perceptibility and stability in optimal parameter estimation in meteorology and oceanography. The first full chemical implementation of the 4D-variational method for reactive atmospheric chemistry initial values jointly with emissions is introduced. Elbern et al. (2000, 2007) took the strong constraint of the diurnal profile shape of emission rates such that their amplitudes and initial values are the only parameters to be optimized by 4D-variational inversion. This strong constraint approach is reasonable because the diurnal evolution sequence of emissions is typically much better known than the absolute amount of daily emissions.

Singular value decomposition (SVD) can help identifying the priorities of observations by detecting the fastest growing uncertainties. Singular vector analysis was firstly introduced to numerical weather prediction by Lorenz (1965), whereas for high-dimensional meteorological models feasibility was demonstrated by Buizza and Palmer 1995. Daescu (2008) introduced a method, exploiting the error covariance sensitivity analysis, to finally assess the data impact on analyses and forecasts. Kang and Xu (2012) applied a 4D-Var system to Burgers' equation to optimize sensor deployment by maximizing observability using a gradient projection approach. Sandu et al. (2013) determined the dominant model singular vectors to identify regions of maximal error growth, which are then candidate locations for optimized sensor placement. In atmospheric chemistry, studies about the importance of observations are still sparse. Khattatov et al. (1999) firstly analysed the uncertainty of a chemical compositions. Liao et al. (2006) focused on the optimal placement of observation locations of the chemical transport model. Starting with a given sensor network, Singh et al. 2013 information theoretical metrics to quantify the value of measurements to reduce the analysis error in the frame of ensemble runs. For accidental releases Abida and Bocquet (2009) sought to reconstruct the plume of emitted compounds by sequentially optimizing observation locations for mobile monitor platforms. However, singular vector analysis and other methods for atmospheric chemistry with emissions are different since emissions plfay a similarly important role in forecast accuracy with initial values. Goris and Elbern (2013) used the singular vector decomposition to determine the sensitivity of the chemical composition to emissions and initial values for a variety of typical chemical scenarios and integration length. This methodology has been generalized for the 3-dimensional EURAD-IM (European Air pollution Dispersion-Inverse Model) and applied to a field campaign with airship borne measurements by Goris and Elbern (2015). While that paper describes an approach to optimize an atmospheric chemistry observation network,

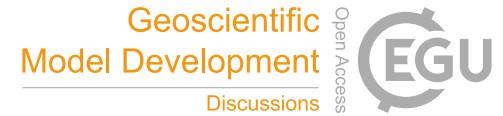


both in terms of individual compounds to be observed with preference and their location, the assessment of the information potential of an established and mainly fixed observation network, like for example the AIRNow AQir Quality Monitor Maps (http://www.airnow.gov/index.cfm?action=airnow.pointmaps) in the US or from the European Environment Agency (http://www.eea.europa.eu/data-and-maps/explore-interactive-maps#c5=&c0=5&b_start=0) needs a different algorithmic ap-

proach. Most measurement devices monitor concentrations hourly or half-hourly.

In practice, the deployment of in situ observations follows mainly legal requirements as manifested in official regulations to monitor concentration threshold violations for public healthcare with emphasis on populated areas. This is in stark contrast to observation network design principles of weather services, which nearly exclusively aspire to comply with data assimilation requirements to optimise initial values for predictions. With the growing importance of earth system modelling and its com-

bination with measurements, existing observation networks need to be validated for forecasting purposes. For this, attention should be paid to the fact that in atmospheric chemistry emission rates are also candidate parameters for optimisation, as they are typically both, insufficiently well known and of high impact on the simulation.

In this paper we introduce the practical implementation of this approach to identify the impacts of the observation networks for controlling tangent linear diffusion models with the Kalman smoother as the appropriate data assimilation method. The

focus is placed on presenting an approach to identify the the spatially resolved potential and limits of measurement networks to optimise chemical states or emission rates or both, by comparing their relative sensitivities. In section 2, we describe an atmospheric transport model extended with emission rates by establishing the dynamic model for emission rates in a novel way. In section 3, based on the Kalman smoother, we derive the theoretical approach to determine the degree of freedom for signal and obtain the equivalence of this approach. In section 5, we develop the ensemble approach to evaluate the degree of

freedom for signal of the model. In section 6, we present the approach to identify the sensitivity of observations by determining the directions of maximum perturbation growth to the initial perturbation and focus on the relationship between section 3 and section 6. In section 7, we extend a 3D advection-diffusion equation with the dynamic model of the emission rate and give several elementary experiments to verify the approaches. In section 8, we conclude the main contributions of this paper and present the further work.

## 2    Atmospheric inverse modelling extended by emission rates

We usually describe the concentration change rate by propagating forward in time the following prognostic atmospheric transport model

$$\frac{dc(t)}{dt} = \mathcal{A}(c) + e(t), \tag{1}$$

where $\mathcal{A}$ is a nonlinear model operator, $c(t)$ and $e(t)$ are the state vector of chemical constituents and emission rates at time $t$,

respectively .

The prior estimate of the state vector of concentrations $c(t)$ is given and denoted by $c_b(t)$, termed the background state. The prior estimate of emission rates, usually taken from emission inventories, is denoted by $e_b(t)$.



Let $\mathbf{A}$ be the tangent linear operator of $\mathcal{A}$, $\delta c(t_0) = c(t_0) - c_b(t_0)$ and $\delta e(t) = e(t) - e_b(t)$. The linear evolution of the perturbation of $c(t)$ follows the tangent linear model as

$$\frac{d\delta c}{dt} = \mathbf{A}\delta c + \delta e(t). \tag{2}$$

By the discretization of the tangent linear model in space, it is straightforward to obtain the linear solution of (2) discretized in space and continuous in time as

$$\delta c(t) = M(t,t_0)\delta c(t_0) + \int_{t_0}^{t} M(t,s)\delta e(s)ds, \tag{3}$$

where $M(\cdot,\cdot)$ is the resolvent obtained from the spacial discretization of $\mathbf{A}$. Without loss of generality, we assume $\delta c(t) \in \mathbf{R}^n$, $\delta e(t) \in \mathbf{R}^n$, where $n$ is the dimension of the partial phase space of concentrations and emission rates. Obviously, $M(\cdot,\cdot) \in \mathbf{R}^{n \times n}$.

In addition, let $y(t)$ be the observation vector of $c(t)$ and define

$$\delta y(t) = y(t) - \mathcal{H}(c_b)(t), \tag{4}$$

where $\delta y(t) \in \mathbf{R}^{m(t)}$, $m(t)$ the dimension of the phase space of observation configurations at time $t$. $\mathcal{H}(t)$ is a nonlinear forward observation operator mapping the model space to the observation space. Then by linearising the nonlinear operator $\mathcal{H}$ as $H$, we present the observation system as

$$\delta y(t) = H(t)\delta c(t) + \nu(t), \tag{5}$$

where the observation error $\nu(t)$ of the Gaussian distribution has zero mean and variance $R(t) \in \mathbf{R}^{m(t) \times m(t)}$.

The Kalman smoother is a recursive estimator to provide the best linear unbiased estimates (BLUE) of the unknown variables using a sequence of observations over time, taking inaccuracies in account, see Gelb (1974). In addition to 4D-Var approaches, Kalman smoothers not only can provide the best linear unbiased estimate by a series of observations over time for the state vector, but also update the forcast error covariances of that estimate. Following the objective of this study, this allows us to determine and balance the most probable parameters of initial values and emission rates from the covariances of estimates. In this paper our problems will be treated by the Kalman smoother from a theoretical viewpoint.

If the initial state of concentration is the only parameter to be optimized, it is feasible to apply the Kalman filter and smoother into the tangent linear model (2) with observations (5) within the time interval $[t_0, t_N]$. However, as mentioned before, in most cases the exact values of emission rates are poorly known. It has been shown by Elbern et al. (2007) that the diurnal profiles of emission rates are better known and hence can be considered as constraints such that the amplitudes of the diurnal emission cycle can be taken as optimization parameters. We first formulate the background evolution of emission rates from time $s$ to $t$ into the dynamic form

$$e_b(t) = M_e(t,s)e_b(s), \quad s \leqslant t, \tag{6}$$



where $e_b(\cdot)$ is a $n$-dimensional vector of which the $i^{th}$ element is denoted by $e_b^i(\cdot)$ and $M_e(t,s)$ is the scaling diagonal matrix defined as

$$M_e(t,s) = \begin{pmatrix} \frac{e_b^1(t)}{e_b^1(s)} & & & \\ & \frac{e_b^2(t)}{e_b^2(s)} & & \\ & & \ddots & \\ & & & \frac{e_b^n(t)}{e_b^n(s)} \end{pmatrix}. \tag{7}$$

In this works we establish the dynamic model of emission rates subject to the constraint

$$5 \quad \delta e(t) = M_e(t,s)\delta e(s), \quad s \leqslant t. \tag{8}$$

It was stated in several studies, for example, Catlin (1989), Gelb (1974), that the estimation of the variable $x$ by fix-interval Kalman smoother generally equals to the conditional expectation based on the observations in the whole time interval, denoted by $\mathbf{E}[x|\{y(t_{obs}), t_{obs} \in [t_0, t_N]\}]$. With the emission model (8), the estimate of $e(t)$ by Kalman smoother on $[t_0, t_N]$ follows the linear property of conditional expectation,

$$10 \quad \mathbf{E}[e(t)|\{y(t_{obs}), t_{obs} \in [t_0, t_N]\}] \tag{9}$$

$$= \mathbf{E}[M_e(t,s)e(s)|\{y(t_{obs}), t_{obs} \in [t_0, t_N]\}] = M_e(t,s)\mathbf{E}[e(s)|\{y(t_{obs}), t_{obs} \in [t_0, t_N]\}].$$

It implies that the BLUEs of emission rates with the dynamic model (8) by Kalman smoother preserve the proportioned diurnal profiles of their backgrounds.

Then we rewrite (3) as

$$15 \quad \delta c(t) = M(t,t_0)\delta c(t_0) + \int_{t_0}^{t} M(t,s)M_e(s,t_0)\delta e(t_0)ds. \tag{10}$$

Further, we obtain the transport model with the state vector extended by emission rates

$$\begin{pmatrix} \delta c(t) \\ \delta e(t) \end{pmatrix} = \begin{pmatrix} M(t,t_0) & \int_{t_0}^{t} M(t,s)M_e(s,t_0)ds \\ 0 & M_e(t,t_0) \end{pmatrix} \begin{pmatrix} \delta c(t_0) \\ \delta e(t_0) \end{pmatrix}. \tag{11}$$

Typically, there is no direct observation for emissions, apart from the flux tower observations used for carbon dioxide, inversion which are not considered here. Therefore, we can reformulate the observation mapping as

$$20 \quad \delta y(t) = (H(t), 0_{n \times n}) \begin{pmatrix} \delta c(t) \\ \delta e(t) \end{pmatrix} + \nu(t), \tag{12}$$

where $0_{n \times n}$ is a $n \times n$ matrix with zero elements.

It is now clear that both concentrations and emission rates are included into the state vector of the homogeneous model (11), It allows us to apply the Kalman smoother in a fixed time interval $[t_0, t_N]$ to optimize both parameters.





In addition, in some practical cases, the initial state and emission rates do not have the same dimension. For example there are more than one different kinds of emissions for one species. This general situation lead us to consider the following model

$$\frac{d\delta c}{dt} = \mathbf{A}\delta c + B(t)\delta e(t). \tag{13}$$

Then combining with (8), we obtain the extended model

$$\begin{pmatrix} \delta c(t) \\ \delta e(t) \end{pmatrix} = \begin{pmatrix} M(t,t_0) & \int_{t_0}^{t} M(t,s)B(s)M_e(s,t_0)ds \\ 0 & M_e(t,t_0) \end{pmatrix} \begin{pmatrix} \delta c(t_0) \\ \delta e(t_0) \end{pmatrix}. \tag{14}$$

However, in order to simply the notation, in this paper we always assume the initial state and emission rates with the same dimension $n$ in section 3 and 5.

## 3  The degree of freedom for signal of discrete-time systems

In some cases where the estimation of both initial state and emission rates can be improved significantly, we can say that the corresponding observation configurations is efficient to optimize both. However, in many cases, the observation configurations are only efficient to the initial state or emission rates, or even to none in case of undue sparseness of measurements. It is usually difficult to quantify the impact of observation configurations on the initial state and emission rates. The lack of knowledge of the efficiency of observations may lead us to give the poor initial guesses, imbalanced and erroneous results. In this section we will introduce the theoretical approach to determine the efficiency of observations by the Kalman smoother in a finite-time interval.

For convenience, we generalize the atmospheric transport model (11) by the following discrete-time linear system on the time interval $[t_0, t_1, \cdots, t_N]$:

$$x(t_{k+1}) = M(t_{k+1}, t_k)x(t_k) + \varepsilon(t_k), \tag{15}$$

$$y(t_k) = H(t_k)x(t_k) + \nu(t_k), \tag{16}$$

where $x(\cdot) \in \mathbf{R}^n$ is the state variable, $y(t_k) \in \mathbf{R}^{m(t_k)}$ is the observation vector at time $t_k$. The model error $\varepsilon(t_k)$ and the observation error $\nu(t_k)$, $k = 1, \cdots, N$ of Gaussian distributions have zero means. The model error covariance matrix is denoted by $Q(t_k)$ and observation error covariance matrix is denoted by $R(t_k)$.

We denote the BLUE of $x(t_i)$ based on $\{y(t_0), \cdots, y(t_k)\}$ by $\hat{x}(t_i|t_k)$, $t_i, t_k \in [t_0, \cdots, t_N]$. Correspondingly, $P(t_i|t_k)$ is defined as the error covariance of $\hat{x}(t_i|t_k)$. It is known that the inverse of the analysis error covariance matrix at initial time, $P^{-1}(t_0|t_N)$, by a fixed-interval Kalman smoother is the optimal Hessian of the underlying cost function of 4D-Var (Li and Navon, 2001). Thus, we have

$$P^{-1}(t_0|t_N) = P^{-1}(t_0|t_{-1}) + \sum_{i=0}^{N} M^{\top}(t_i, t_0)H^{\top}(t_i)R^{-1}(t_i)H(t_i)M(t_i, t_0). \tag{17}$$





It implies the equivalence between Kalman smoother and 4D-Var method for linear models (Li and Navon, 2001) and guarantees that the following approach in this paper is available for the data assimilation based on 4D-Var method.

It is clear that (17) comprises the information of the initial condition, model evolution, observation configurations and errors over the entire time interval $[t_0, \cdots, t_N]$. At the same time, it is independent of any specific data and state vector, apart from the reference model evolution $M(\cdot, \cdot)$ needed for the linearisation, as well as the observation operator $H(\cdot)$. Actually, if we define

$$
\mathcal{G} = \begin{pmatrix} H(t_0)M(t_0,t_0) \\ H(t_1)M(t_1,t_0) \\ \vdots \\ H(t_N)M(t_N,t_0) \end{pmatrix}, \quad \mathcal{R}^{-1} = \begin{pmatrix} R^{-1}(t_0) & & & \\ & R^{-1}(t_1) & & \\ & & \ddots & \\ & & & R^{-1}(t_N) \end{pmatrix},
\tag{18}
$$

we can rewrite (17) as

$$
P^{-1}(t_0|t_N) = P^{-1}(t_0|t_{-1}) + \mathcal{G}^\top \mathcal{R}^{-1} \mathcal{G},
\tag{19}
$$

where $\mathcal{G}^\top R^{-1} \mathcal{G}$ is the observability Gramian with respect to $\mathcal{R}^{-1}$ in control theory (Brockett, 1994). It represents the observation capacity of the observation networks with respect to the model.

Though (19) meets the demand to represent the estimate covariance by all available information before starting the data assimilation procedure, it cannot be applied directly to evaluate the potential improvement of the estimate by the Kalman smoother due to the lack of clear statistical significance of the inverse of a covariance matrix. Thus, aspiring a means to compare the capacity of the observation to improve the estimation of initial values and emission rates in a scaled way, we define a matrix $\tilde{P}$ as

$$
\tilde{P} = P^{-\frac{1}{2}}(t_0|t_{-1})(P(t_0|t_{-1}) - P(t_0|t_N))P^{-\frac{1}{2}}(t_0|t_{-1}) = I - P^{-\frac{1}{2}}(t_0|t_{-1})P(t_0|t_N)P^{-\frac{1}{2}}(t_0|t_{-1}),
\tag{20}
$$

where $I$ is the identity matrix.

The matrix $\tilde{P}$ is a normalized matrix of the difference between the background forecast error covariance matrix $P(t_0|t_{-1})$ and the analysis error covariance matrix $P(t_0|t_N)$ from the Kalman smoother. Especially, $P^{-\frac{1}{2}}(t_0|t_{-1})P(t_0|t_N)P^{-\frac{1}{2}}(t_0|t_{-1})$ can be understood as the covariance matrix from the fixed-interval Kalman smoother normalized by the initial variance. For any nonnegative matrices $A, B \in \mathbf{R}^{n \times n}$, we define $A \prec (\preccurlyeq) B$ if $x^\top A x < (\leqslant) x^\top B x$ for any $x \neq 0_{n \times 1}$. Thus, the symmetry of (20) guarantees $\tilde{P}$ to be nonnegative-definite. In fact,

$$
0 \preccurlyeq P^{-\frac{1}{2}}(t_0|t_{-1})(P(t_0|t_{-1}) - P(t_0|t_N))P^{-\frac{1}{2}}(t_0|t_{-1}) \prec I,
\tag{21}
$$

where the left equality holds for the situation that there is no observation within $[t_0, \cdots, t_N]$. Further, (21) implies that its singular values are bounded by $n$ since their sum is less than $n$, that is the trace of $I$.

Since $P(t_0|t_N)$ is unknown prior to the data assimilation procedure, we use (19) to rewrite $\tilde{P}$ as

$$
\tilde{P} = P^{-\frac{1}{2}}(t_0|t_{-1})(P(t_0|t_{-1}) - P(t_0|t_N))P^{-\frac{1}{2}}(t_0|t_{-1})
\tag{22}
$$





$$= P^{-\frac{1}{2}}(t_0|t_{-1})(P(t_0|t_{-1}) - (P^{-1}(t_0|t_{-1}) + \mathcal{G}^\top \mathcal{R}^{-1} \mathcal{G})^{-1})P^{-\frac{1}{2}}(t_0|t_{-1})$$

$$= I - P^{-\frac{1}{2}}(t_0|t_{-1})(P^{-1}(t_0|t_{-1}) + \mathcal{G}^\top \mathcal{R}^{-1} \mathcal{G})^{-1}P^{-\frac{1}{2}}(t_0|t_{-1})$$

$$= I - (I + P^{\frac{1}{2}}(t_0|t_{-1})\mathcal{G}^\top \mathcal{R}^{-1} \mathcal{G} P^{\frac{1}{2}}(t_0|t_{-1}))^{-1}.$$

It is worth noting that in (22)

$$I + P^{\frac{1}{2}}(t_0|t_{-1})\mathcal{G}^\top \mathcal{R}^{-1} \mathcal{G} P^{\frac{1}{2}}(t_0|t_{-1}) \tag{23}$$

is always invertible even if the observation Gramian $\mathcal{G}^\top \mathcal{G}$ is not full-rank. Thus, $\tilde{P}$ is well-defined for all models with invertible initial covariance and observation systems with invertible error covariances within assimilation window $t_0$ to $t_N$. Due to the high computational costs of (22), we apply the singular value decomposition to

$$P^{\frac{1}{2}}(t_0|t_{-1})\mathcal{G}^\top \mathcal{R}^{-\frac{1}{2}} = VSU^\top, \tag{24}$$

where $V$ and $U$ are unitary matrices consisting of the left and right singular vectors, respectively, while $S$ is the rectangular diagonal matrix consisting of the singular values.

Then, (22) can be simplified as

$$\begin{aligned}
\tilde{P} &= I - (I + P^{\frac{1}{2}}(t_0|t_{-1})\mathcal{G}^\top \mathcal{R}^{-1} \mathcal{G} P^{\frac{1}{2}}(t_0|t_{-1}))^{-1} \\
&= I - (I + VSS^\top V^\top)^{-1} \\
&= VV^\top - (VV^\top + VSS^\top V^\top)^{-1} \\
&= VV^\top - (V(I + SS^\top)V^\top)^{-1} \\
&= V(I - (I + SS^\top)^{-1})V^\top \\
&= \sum_{i=1}^{r} \frac{s_i^2}{1 + s_i^2} v_i v_i^\top,
\end{aligned}$$

where $r$ is the rank of (22) and $v_i$ is the $i^{th}$ left singular vector in $V$ related to the singular value $s_i$, which is the $i^{th}$ element on the diagonal of $S$.

It is clear that the sum of the diagonal entries of $\tilde{P}$ can be considered to evaluate the total improvement of the estimate, 1-norm, also named as nuclear norm, is appropriately taken as the metric, which is defined as

$$\|A\|_1 = \operatorname{tr}(\sqrt{A^\top A}), \tag{25}$$

where $A$ is any matrix and $\operatorname{tr}(\cdot)$ denotes the trace of the matrix.

From (25), we obtain

$$\|\tilde{P}\|_1 = \operatorname{tr}(\tilde{P}) = \sum_{i=1}^{r} \frac{s_i^2}{1 + s_i^2}. \tag{26}$$

According to Rodgers (2000), it is called as the degree of freedom for signal (DFS).





As we mentioned before, $\|\tilde{P}\|_1 < \|I\|_1 = n$. Here $n$ is considered as the total improvement if the system is definitely observed. Thus, if we consider the ratio

$$\tilde{p} = \frac{\|\tilde{P}\|_1}{\|I\|_1} = \frac{\|\tilde{P}\|_1}{n} \in [0,1), \tag{27}$$

the percentage of the total improvement of the model is obtained, which is called the *relative degree of freedom for signal*.

5    In order to get a deeper insight into the capacity of the observation networks to improve the estimation of all model states, we consider the corresponding value in the diagonal of $\tilde{P}$ as *the contribution of the degree of freedom for signal*. Denote the $j^{th}$ element on the diagonal of $\tilde{P}$ by $\tilde{P}_j$, the contribution of the $j^{th}$ element of $x(t_0)$ to the degree of freedom for signal can be expressed, from (25), as

$$\tilde{P}_j = \sum_{i=1}^{r} \frac{s_i^2}{1+s_i^2}(v_{ij})^2, \tag{28}$$

10    where $v_{ij}$ is the $j^{th}$ element of $v_i$.

## 4    The degrees of freedom for signal of the initial state and emission rates

Equation (25) enable us to discriminate the DFS contributed to different optimisation parameters, that is emission rates and initial values. Without loss of generality, we simply assume the original state $c \in \mathbf{R}^n$ and emission rates $e \in \mathbf{R}^n$. We divide (22) into the following block matrix according to the dimension of $c$ and $e$

$$15 \quad \tilde{P} = \begin{pmatrix} \tilde{P}^c & \tilde{P}^{ce} \\ \tilde{P}^{ec} & \tilde{P}^e \end{pmatrix} = \sum_{i=1}^{2n} \frac{s_i^2}{1+s_i^2} \begin{pmatrix} v_i^c \\ v_i^e \end{pmatrix} (v_i^{c\top}, v_i^{e\top}) \in \mathbf{R}^{2n \times 2n}, \tag{29}$$

where $(v_i^{c\top}, v_i^{e\top})^\top = v_i$.

It is easy to see that

$$\tilde{P}^c = \sum_{i=1}^{2n} \frac{s_i^2}{1+s_i^2} v_i^c v_i^{c\top}, \quad \tilde{P}^e = \sum_{i=1}^{2n} \frac{s_i^2}{1+s_i^2} v_i^e v_i^{e\top}. \tag{30}$$

Further, the degree of freedom for signal of $j^{th}$ element in $c(t_0)$ and $e(t_0)$ are given by

$$20 \quad \tilde{P}_j^c = \sum_{i=1}^{2n} \frac{s_i^2}{1+s_i^2}(v_{ij}^c)^2, \quad \tilde{P}_j^e = \sum_{i=1}^{2n} \frac{s_i^2}{1+s_i^2}(v_{ij}^e)^2, \tag{31}$$

where $v_{ij}^c$ and $v_{ij}^e$ are the $j^{th}$ elements of $v_i^c$ and $v_i^e$ respectively.

Moreover, the degree of freedom for signal of concentration $\|\tilde{P}^c\|_1$ and emission rates $\|\tilde{P}^e\|_1$ are caculated by

$$\|\tilde{P}^c\|_1 = \sum_{i=1}^{2n} \frac{s_i^2}{1+s_i^2}\mathrm{tr}(v_i^c v_i^{c\top}), \quad \|\tilde{P}^e\|_1 = \sum_{i=1}^{2n} \frac{s_i^2}{1+s_i^2}\mathrm{tr}(v_i^e v_i^{e\top}). \tag{32}$$





It is worth noticing that

$$\tilde{P}^c = (P^c(t_0|t_{-1}))^{-\frac{1}{2}}(P^c(t_0|t_{-1}) - P^c(t_0|t_N))(P^c(t_0|t_{-1}))^{-\frac{1}{2}} \tag{33}$$

$$\tilde{P}^e = (P^e(t_0|t_{-1}))^{-\frac{1}{2}}(P^e(t_0|t_{-1}) - P^e(t_0|t_N))(P^e(t_0|t_{-1}))^{-\frac{1}{2}} \tag{34}$$

if and only if there is no prior correlation between the initial concentration and emission rates. In this case $P^{ce}(t_0|t_{-1}) = 0_{n \times n}$,

the corresponding relative degrees of freedom for signal of concentration and emission rates are defined as

$$\tilde{p}^c = \frac{\|\tilde{P}^c\|_1}{n}, \quad \tilde{p}^e = \frac{\|\tilde{P}^e\|_1}{n}. \tag{35}$$

From (27), it is obvious that $\tilde{p}^c \in [0,1)$ and $\tilde{p}^e \in [0,1)$ can be considered as the percentages of the relative improvements of concentration and emission rates, respectively. However, efficient observation networks probably lead to both of them are close to 1 such that

$$\frac{\|\tilde{P}^c\|_1}{n} + \frac{\|\tilde{P}^e\|_1}{n} > 1. \tag{36}$$

It indicates the normalization of $\tilde{P}$ is only with respect to the extended covariance matrix rather than specified to the state $c$ and emission rates $e$. The relative degree of freedom for signal cannot serve our objective to distinguish the observability of concentration and emission rates. However, by observing the block form of $\tilde{P}$, we have

$$\|\tilde{P}^c\|_1 + \|\tilde{P}^e\|_1 = \|\tilde{P}\|_1. \tag{37}$$

Thus, in order to compare the improvements of the concentration and emission rates, we define *relative ratio of the degree of freedom for signal* for concentrations or emission rates as

$$\tilde{p}^c = \frac{\|\tilde{P}^c\|_1}{\|\tilde{P}\|_1}, \quad \tilde{p}^e = \frac{\|\tilde{P}^e\|_1}{\|\tilde{P}\|_1}, \quad \tilde{p}^e + \tilde{p}^c \equiv 1. \tag{38}$$

If the degree or relative degree of freedom for signal of the observation network and assimilation window is almost zero, an improvement cannot be expected. In contrast, $\{\tilde{P}^c_j\}_{j=1}^n$ and $\{\tilde{P}^e_j\}_{j=1}^n$, which show the improvement of each parameter $j$

of concentrations and emission rates respectively, can help us determining which parameters can be optimized by the existing observation configurations. Furthermore, comparing $\tilde{p}^c$ with $\tilde{p}^e$, we can conclude that the estimate of the one with the larger relative ratio of freedom for signal can be improved more efficiently by the existing observation configurations than the other. In other words, if $\tilde{p}^c > \tilde{p}^e$, the existing observation configurations are more efficient to the initial values of concentrations. Conversely, if $\tilde{p}^c < \tilde{p}^e$, the observation configurations can improve the estimate of emission rates more. According to $\tilde{p}^c$ and

$\tilde{p}^e$, the "weights" between the concentrations and emission rates can be identified quantitatively. In a data assimilation context, where observations are in a weighted relation to the background, the BLUE favours those parameters with higher observation efficiency.

The special case that $\tilde{p}^e$ is very close to zero implies that observation network is nearly "blind" for emission rate optimisation.



## 5   The ensemble approach for determing the DFS

The ensemble Kalman smoother (EnKS), as a Monte Carlo implementation derived from the Kalman smoother, is suitable for problems with a large number of control variables and is a frequently applied tool in the field of data assimilation (Evensen, 2009). In this section we will introduce the ensemble case of the determination of DFS.

For the discrete-time system (15), we denote the ensemble samples of $\hat{x}(t_i|t_j)$ by $\hat{x}_k(t_i|t_j)$, $i,j = 1, \cdots, N$, $k = 1, \cdots, q$, where $q$ is the number of ensemble members.

Correspondingly, the ensemble means of $\hat{x}(t_i|t_j)$ is given by

$$\bar{x}(t_i|t_j) = \frac{1}{q} X(t_i|t_j) \mathbf{1}_{q \times 1}, \tag{39}$$

where $X(t_i|t_j) = (\hat{x}_1(t_i|t_j), \hat{x}_2(t_i|t_j), \cdots, \hat{x}_q(t_i|t_j))$ is the $n \times q$ ensemble matrix, $\mathbf{1}_{i \times j}$ is a $i \times j$ matrix of which each element is equal to $1$.

We calculate the ensemble forecast and analysis covariances as

$$\bar{P}(t_i|t_j) = \frac{1}{q-1} \tilde{X}(t_i|t_j) \tilde{X}^{\top}(t_i|t_j), \tag{40}$$

where $\tilde{X}(t_i|t_j) = X(t_i|t_j) - \frac{1}{q} X(t_i|t_j) \mathbf{1}_{q \times q}$ is the related perturbation matrix. We define the ensemble observation configurations in the entire assimilation window as

$$y_k^f = \mathcal{G} \hat{x}_k(t_0|t_{-1}), \quad k = 1, \cdots, q. \tag{41}$$

Further, the ensemble mean and the forecast error covariance matrix of the ensemble observation configurations are given by

$$\bar{y}^f = \frac{1}{q} \sum_{k=1}^{q} y_k^f, \quad \bar{P}_{yy}^f = \frac{1}{q-1} \sum_{k=1}^{q} (\hat{y}_k^f - \bar{y}^f)(\hat{y}_k^f - \bar{y}^f)^{\top} = \mathcal{G} \bar{P}(t_0|t_{-1}) \mathcal{G}^{\top}. \tag{42}$$

Similarly, we denote the ensemble covariance between the initial states and the forecasted observations by

$$\bar{P}_{xy}^f = \frac{1}{q-1} \sum_{k=1}^{q} (\hat{x}_k(t_0|t_{-1}) - \bar{x}(t_0|t_{-1}))(\hat{y}_k^f - \bar{y}^f)^{\top} = \bar{P}(t_0|t_{-1}) \mathcal{G}^{\top}. \tag{43}$$

Furthermore, defining the ensemble observations as

$$\hat{y}_k(t_i) = y(t_i) + \nu_k(t_i), \quad k = 1, \cdots, q, \quad i = 1, \cdots, N, \tag{44}$$

we assume $\bar{\nu}(t_i) = \frac{1}{q} \sum_{k=1}^{q} \nu_k(t_i) = 0$, $\bar{R}(t_i) = \frac{1}{q-1} \sum_{k=1}^{q} \nu_k(t_i) \nu_k^{\top}(t_i)$ and $\bar{\mathcal{R}}^{-1}$ is the block diagonal matrix with the diagonal $(\bar{R}^{-1}(t_0), \cdots, \bar{R}^{-1}(t_N))$.

It is shown by Evensen (2009) that the ensemble forecast and analysis covariances have the same form with the covariances in the standard Kalman filter. However, the ensemble size $q$ is significantly less than the dimension of the model $n$ in pratical applications. It causes that the initial ensemble covariance $\bar{P}(t_0|t_{-1})$ is not invertible. In this case, the pseudo inverse is a





widely used alternative of the inverse of a matrix, due to its best fitness and uniqueness. We denote the pseudo inverse of a matrix $A$ by $A^\dagger$. Then for the initial ensemble covariance

$$\bar{P}(t_0|t_{-1}) = \frac{1}{q-1}\tilde{X}(t_0|t_{-1})\tilde{X}^\top(t_0|t_{-1}),$$ (45)

we apply the singular value decomposition to

5 $$\frac{1}{\sqrt{q-1}}\tilde{X}(t_0|t_{-1}) = V_0 S_0 U_0^\top,$$ (46)

where $V_0 \in \mathbf{R}^{n\times n}$ and $U_0 \in \mathbf{R}^{q\times q}$ consist of the left and right singular vectors respectively, and $S_0 \in \mathbf{R}^{n\times q}$ is a rectangular diagonal matrix with singular values $\{s_{0i}|s_{0i} \geqslant 0\}_{i=1}^q$ on its diagonal. Thus,

$$\bar{P}(t_0|t_{-1}) = V_0 S_0 U_0^\top U_0 S_0^\top V_0^\top = V_0 S_0 S_0^\top V_0^\top = V_0 \hat{S}_0^2 V_0^\top,$$ (47)

where $\hat{S}_0^2 = S_0 S_0^\top \in \mathbf{R}^{n\times n}$ is a block diagonal matrix with the diagonal $(s_{01}^2,\cdots,s_{0r_0}^2, 0_{1\times(n-r_0)})$, $r_0$ is the rank of $S_0$. Hence,

we find a pseudo inverse

$$\bar{P}^{\dagger\frac{1}{2}}(t_0|t_{-1}) = V_0 \hat{S}_0^\dagger V_0^\top,$$ (48)

where $\hat{S}_0^\dagger$ is the pseudo inverse of $\hat{S}_0$ with the diagonal $(1/s_{01},\cdots,1/s_{0r_0}, 0_{1\times(n-r_0)})$. Analog to (20), we define

$$\bar{P} = \bar{P}^{\dagger\frac{1}{2}}(t_0|t_{-1})(\bar{P}(t_0|t_{-1}) - \bar{P}(t_0|t_N))\bar{P}^{\dagger\frac{1}{2}}(t_0|t_{-1}).$$ (49)

Likewise, corresponding to (16), we present the observation system in the entire time interval as

$$y = \mathcal{G}x(t_0) + \nu,$$ (50)

where $y = (y^\top(t_0),\cdots,y^\top(t_N))^\top$, $\nu = (\nu^\top(t_0),\cdots,\nu^\top(t_N))^\top$ and $\mathcal{G}$ as the observation configuration for $x(t_0)$. Then, for the analysis error covariance matrix, we obtain

$$\begin{aligned}
&\bar{P}(t_0|t_N) \\
=\ & \bar{P}(t_0|t_{-1}) - \bar{P}(t_0|t_{-1})\mathcal{G}^\top(\mathcal{G}\bar{P}(t_0|t_{-1})\mathcal{G}^\top + \mathcal{R})^{-1}\mathcal{G}\bar{P}(t_0|t_{-1}) \\
=\ & \bar{P}(t_0|t_{-1}) - \bar{P}(t_0|t_{-1})\mathcal{G}^\top\mathcal{R}^{-\frac{1}{2}}(I + \mathcal{R}^{-\frac{1}{2}}\mathcal{G}\bar{P}(t_0|t_{-1})\mathcal{G}^\top\mathcal{R}^{-\frac{1}{2}})^{-1}\mathcal{R}^{-\frac{1}{2}}\mathcal{G}\bar{P}(t_0|t_{-1}) \\
=\ & \bar{P}(t_0|t_{-1}) - \bar{P}_{xy}^f\mathcal{R}^{-\frac{1}{2}}(I + \mathcal{R}^{-\frac{1}{2}}\bar{P}_{yy}^f\mathcal{R}^{-\frac{1}{2}})^{-1}\mathcal{R}^{-\frac{1}{2}}(\bar{P}_{xy}^f)^\top.
\end{aligned}$$ (51)

Further, analog to (25), we obtain

$$\begin{aligned}
\bar{P} =\ & \bar{P}^{\dagger\frac{1}{2}}(t_0|t_{-1})(\bar{P}(t_0|t_{-1}) - \bar{P}(t_0|t_N))\bar{P}^{\dagger\frac{1}{2}}(t_0|t_{-1}) \\
=\ & \bar{P}^{\dagger\frac{1}{2}}(t_0|t_{-1})\bar{P}_{xy}^f\mathcal{R}^{-\frac{1}{2}}(I + \mathcal{R}^{-\frac{1}{2}}\bar{P}_{yy}^f\mathcal{R}^{-\frac{1}{2}})^{-1}\mathcal{R}^{-\frac{1}{2}}(\bar{P}_{xy}^f)^\top\bar{P}^{\dagger\frac{1}{2}}(t_0|t_{-1}).
\end{aligned}$$ (52)

Let $\sum_{i=1}^N m(t_i) = m$ be the number of observations availabel within the assimilation window. To proceed with (52), we apply again the singular value decomposition to find,

$$\bar{P}^{\dagger\frac{1}{2}}(t_0|t_{-1})\bar{P}_{xy}^f\mathcal{R}^{-\frac{1}{2}} = VSU^\top \in \mathbf{R}^{n\times m},$$ (53)



where $U \in \mathbf{R}^{m \times m}$ and $V \in \mathbf{R}^{n \times n}$ consists of the eigenvectors of $\mathcal{R}^{-\frac{1}{2}}\mathcal{G}\bar{P}(t_0|t_{-1})\mathcal{G}^{\top}\mathcal{R}^{-\frac{1}{2}}$ and $\bar{P}^{\frac{1}{2}}(t_0|t_{-1})\mathcal{G}^{\top}\mathcal{R}^{-1}\mathcal{G}\bar{P}^{\frac{1}{2}}(t_0|t_{-1})$, respectively. $S \in \mathbf{R}^{n \times m}$ consists of the singular values on its diagonal.

We denote the rank of (53) by $r$. Then, $\bar{P}$ can be rewritten as

$$
\begin{aligned}
\bar{P} &= \bar{P}^{\dagger\frac{1}{2}}(t_0|t_{-1})(\bar{P}(t_0|t_{-1}) - \bar{P}(t_0|t_N))\bar{P}^{\dagger\frac{1}{2}}(t_0|t_{-1}) \\
&= VS^{\top}U^{\top}(UU^{\top} + U(SS^{\top})U^{\top})^{-1}USV^{\top} \\
&= VS^{\top}(I + S^{\top}S)^{-1}SV^{\top} \\
&= \sum_{i=1}^{r} \frac{s_i^2}{1 + s_i^2}v_i v_i^{\top}.
\end{aligned}
\tag{54}
$$

We observe that (54) and (25) have a similar form. By virtue of

$$
\bar{P}^{\dagger\frac{1}{2}}(t_0|t_{-1})\bar{P}_{xy}^f\bar{\mathcal{R}}^{-\frac{1}{2}} = \bar{P}^{\frac{1}{2}}(t_0|t_{-1})\mathcal{G}^{\top}\bar{\mathcal{R}}^{-\frac{1}{2}},
\tag{55}
$$

the final results of (25) and (54) are equivalent. However, compared with $P^{\frac{1}{2}}(t_0|t_{-1})\mathcal{G}^{\top}\mathcal{R}^{-\frac{1}{2}}$, the ensemble expression $\bar{P}^{\dagger\frac{1}{2}}(t_0|t_{-1})\bar{P}_{xy}^f\bar{\mathcal{R}}^{-\frac{1}{2}}$ processes the absolute advantage that in the calculation of $\bar{P}_{xy}^f$, we do not need the explicit form of $\mathcal{G}$. It allows us to code it line by line such that our approach is computationally more efficient.

Analog to the standard case, we can similarly define *the ensemble degree of freedom for signal* (EnDFS) as $\|\bar{P}\|_1$ and consider each element on the diagonal of $\bar{P}$ as *the contribution to EnDFS* of the corresponding model state.

Since $\bar{P}(t_0|t_{-1})$ is typically not full rank,

$$
\begin{aligned}
&\bar{P}^{\dagger\frac{1}{2}}(t_0|t_{-1})(\bar{P}(t_0|t_{-1}) - \bar{P}(t_0|t_N))\bar{P}^{\dagger\frac{1}{2}}(t_0|t_{-1}) \\
&= \bar{P}^{\dagger\frac{1}{2}}(t_0|t_{-1})\bar{P}(t_0|t_{-1})\bar{P}^{\dagger\frac{1}{2}}(t_0|t_{-1}) - \bar{P}^{\dagger\frac{1}{2}}(t_0|t_{-1})\bar{P}(t_0|t_N)\bar{P}^{\dagger\frac{1}{2}}(t_0|t_{-1}) \\
&= V_0\hat{S}_0^{\dagger}V_0^{\top}(V_0\hat{S}_0^2 V_0^{\top})V_0\hat{S}_0^{\dagger}V_0^{\top} - \bar{P}^{\dagger\frac{1}{2}}(t_0|t_{-1})\bar{P}(t_0|t_N)\bar{P}^{\dagger\frac{1}{2}}(t_0|t_{-1}) \\
&= V_0 I_{r_0}V_0^{\top} - \bar{P}^{\dagger\frac{1}{2}}(t_0|t_{-1})\bar{P}(t_0|t_N)\bar{P}^{\dagger\frac{1}{2}}(t_0|t_{-1}),
\end{aligned}
\tag{56}
$$

where $I_{r_0}$ is the diagonal matrix with the diagonal $(\mathbf{1}_{1 \times r_0}, 0_{1 \times (n-r_0)})$.

It is clear from (51) that $\bar{P}^{\dagger\frac{1}{2}}(t_0|t_{-1})\bar{P}(t_0|t_N)\bar{P}^{\dagger\frac{1}{2}}(t_0|t_{-1})$ is still nonnegative definite and $0_{n \times n} \preccurlyeq \bar{P} \prec I_{r_0}$.

Thus, the *ensemble relative degree of freedom for signal*(EnRDFS) is defined by

$$
\bar{p} = \frac{\|\bar{P}\|_1}{\|I_{r_0}\|_1} = \frac{\|\bar{P}\|_1}{r_0} \in [0,1).
\tag{57}
$$

For the distinction of the improvements for concentrations and emission rates, the *ensemble relative ratios of DFS* remain

$$
\bar{p}^c = \frac{\|\bar{P}^c\|_1}{\|\bar{P}\|_1}, \quad \bar{p}^e = \frac{\|\bar{P}^e\|_1}{\|\bar{P}\|_1}.
\tag{58}
$$

If we further consider the nonlinear dynamic model, we can renew the definition of the forecasting observation configurations as

$$
y_k^f = \mathcal{G}(\hat{x}_k(t_0|t_{-1})), \quad k = 1,\cdots,q,
\tag{59}
$$



such that it can follow the nonlinear model, where $\mathcal{G}$ is again a combined model-observation nonlinear operator.

Correspondingly, its ensemble mean and covariance are

$$\bar{y}^f = \frac{1}{q}\sum_{k=1}^{q} y_k^f, \quad \bar{P}_{yy}^f = \frac{1}{q-1}\sum_{k=1}^{q}(\hat{y}_k^f - \bar{y}^f)(\hat{y}_k^f - \bar{y}^f)^\top. \tag{60}$$

## 6 Sensitivity of observation networks

The above discussion about the observation efficiency aims to evaluate a predefined measurement network on its potential to analyse initial values and emission rates simultaneously. In this section, we will introduce the singular vector approach to identify the sensitive directions of observation networks to initial values and emission rates and show the association between the efficiency and sensitivity of observation networks.

We denote $\delta x(t_0) = x(t_0) - \hat{x}(t_0) \in \mathbf{R}^{n\times 1}$, $\hat{x}(t_0)$ is any estimate of $x(t_0)$ and consider the discrete-time linear system in $[t_0,\cdots,t_N]$

$$\delta x(t_{k+1}) = M(t_{k+1},t_k)\delta x(t_k) \tag{61}$$

with the observation configuration

$$\delta y_c(t_k) = H(t_k)\delta x(t_k), \quad \delta y_c(t_k) \in \mathbf{R}^{m(t)\times 1}. \tag{62}$$

Then we define the magnitude of the perturbation of the initial state by the norm in the state space with respect to a positive definite matrix $W_0$ typically the forecast error covariance matrix.

$$\|\delta x(t_0)\|_{W_0}^2 = \langle \delta x(t_0), W_0\delta x(t_0)\rangle. \tag{63}$$

Likewise, we define the magnitude of the related observations perturbation in the time interval $[t_0,\cdots,t_N]$ by the norm with respect to a sequence of positive definite matrices $\{W(t_k)\}_{k=1}^N$

$$\|\delta y_c\|_{\{W(t_k)\}}^2 = \sum_{k=0}^{N}\langle \delta y_c(t_k), W(t_k)\delta y_c(t_k)\rangle, \tag{64}$$

where $\delta y_c = (\delta y_c^\top(t_0),\cdots,\delta y_c^\top(t_N))^\top$.

In order to find the direction of observation configuration which can minimize the perturbation of the initial states, the ratio

$$\frac{\|\delta x(t_0)\|_{W_0}^2}{\|\delta y_c\|_{\{W(t_k)\}}^2}, \quad \delta y \neq 0_{m\times 1}. \tag{65}$$

should be minimized. It is equivalent to maximize the ratio of the magnitude of observation perturbation and the initial perturbation

$$\frac{\|\delta y_c\|_{\{W(t_k)\}}^2}{\|\delta x(t_0)\|_{W_0}^2}, \quad \delta x(t_0) \neq 0_{n\times 1}. \tag{66}$$





Thus, we define the measure the perturbation growth as

$$g^2 = \frac{\|\delta y_c\|^2_{\{W(t_k)\}}}{\|\delta x(t_0)\|^2_{W_0}} \tag{67}$$

$$= \sum_{k=0}^{N} \frac{\langle \delta y_c(t_k), W(t_k)\delta y_c(t_k)\rangle}{\langle \delta x(t_0), W_0\delta x(t_0)\rangle}$$

$$= \sum_{k=0}^{N} \frac{\langle H(t_k)\delta x(t_k), W(t_k)H(t_k)\delta x(t_k)\rangle}{\langle \delta x(t_0), W_0\delta x(t_0)\rangle}$$

$$= \sum_{k=0}^{N} \frac{\langle \delta x(t_k), H(t_k)^\top W(t_k)H(t_k)\delta x(t_k)\rangle}{\langle \delta x(t_0), W_0\delta x(t_0)\rangle}$$

$$= \sum_{k=0}^{N} \frac{\langle \delta x(t_0), M(t_k,t_0)^\top H(t_k)W(t_k)H(t_k)M(t_k,t_0)\delta x(t_0)\rangle}{\langle \delta x(t_0), W_0\delta x(t_0)\rangle}$$

$$= \frac{\langle \delta x(t_0), \sum_{k=0}^{N} M(t_k,t_0)^\top H(t_k)^\top W(t_k)H(t_k)M(t_k,t_0)\delta x(t_0)\rangle}{\langle \delta x(t_0), W_0\delta x(t_0)\rangle}$$

$$= \frac{\langle \delta x(t_0), \mathcal{G}^\top \mathcal{W}\mathcal{G}\delta x(t_0)\rangle}{\langle \delta x(t_0), W_0\delta x(t_0)\rangle}, \quad \delta x(t_0) \neq 0, \tag{68}$$

where $\mathcal{G}$ has the same definition as given in Section 3, yet

$$\mathcal{W} = \begin{pmatrix} W(t_0) & & \\ & \ddots & \\ & & W(t_N) \end{pmatrix}. \tag{69}$$

Then we arrive at the eigenvalue problems (Liao et al., 2006)

$$W_0^{-\frac{1}{2}}\mathcal{G}^\top \mathcal{W}\mathcal{G}W_0^{-\frac{1}{2}}v_k = s_k^2 v_k, \quad W^{\frac{1}{2}}\mathcal{G}W_0^{-1}\mathcal{G}^\top W^{\frac{1}{2}}u_k = s_k^2 u_k, \tag{70}$$

where $s_1 \geqslant s_2 \geqslant \cdots \geqslant s_n \geqslant 0$, $\{v_k\}_{i=1}^n$ and $\{u_k\}_{i=1}^n$ are the corresponding orthogonal singular vectors. Then, $\max_{\delta x(t_0)\neq 0} g^2 = s_1^2$.

Especially, if the perturbation norms are provided by the choice $W_0 = P^{-1}(t_0|t_{-1})$ and $\mathcal{W} = \mathcal{R}^{-1}$,

$$g^2 = \frac{\langle \delta x(t_0), \mathcal{G}^\top \mathcal{R}^{-1}\mathcal{G}\delta x(t_0)\rangle}{\langle \delta x(t_0), P^{-1}(t_0|t_{-1})\delta x(t_0)\rangle}, \quad \delta x(t_0) \neq 0. \tag{71}$$

We need to search the directions of

$$P^{\frac{1}{2}}(t_0|t_{-1})\mathcal{G}^\top \mathcal{R}^{-1}\mathcal{G}P^{\frac{1}{2}}(t_0|t_{-1})v_k = s_k^2 v_k; \tag{72}$$

$$\mathcal{R}^{-\frac{1}{2}}\mathcal{G}P(t_0|t_{-1})\mathcal{G}^\top \mathcal{R}^{-\frac{1}{2}}u_k = s_k^2 u_k, \quad k = 1, \cdots, n. \tag{73}$$

The singular value $s_k$ shows the amplification of the impact of the initial state to the observation configurations during the entire time interval. The associated singular vector in the state space $v_k$ is the direction of $k^{\text{th}}$ growth of the perturbation of observations evolved from the initial perturbation. We compare the sensitivity analysis with the analysis in Section 3. Since



$\frac{s_k^2}{1+s_k^2}$ are decreasing with the decrease of $s_k$, $k = 1, \cdots, n$, it is clear that $v_k$ is also the k$^{\text{th}}$ direction which maximizes the relative improvement of estimates based on Kalman smoother. It indicates that the states contributing to DFS more are the same with the states most sensitive to the observation networks. Besides, the leading singular value $s_1$ is related to the operator norm of $\tilde{P}$ as

$$\|\tilde{P}\| = \max_{\|x\|=1} \|\tilde{P}x\| = \frac{s_1^2}{1+s_1^2}, \tag{74}$$

which implies the upper boundedness of the relative improvement covariance. It gives us an access to approximate and target the sensitive parameters or areas with the metric of the leading singular vectors weighted by the corresponding singular values.

Moreover, due to the homogeneity of the atmospheric transport model state vector extended with emissions, the above sensitivity analysis can be easily applied by dividing singular vectors into the block form according to the dimensions of the initial state and emissions. The corresponding block parts of different singular vectors indicate the different sensitive directions of the initial state and emissions and allow for this relative quantification. Correspondingly, it can be judged to what extend application is feasible with the analysed observation configuration.

## 7  Example

We consider a linear advection-diffusion model with Dirichlet horizontal boundary condition and Neumann boundary condition in the vertical direction on the domain $[0, 14] \times [0, 14] \times [0, 4]$,

$$\frac{\partial \delta c}{\partial t} = -v_x \frac{\partial \delta c}{\partial x} - v_y \frac{\partial \delta c}{\partial y} + \frac{\partial}{\partial z}\left(K(z)\frac{\partial \delta c}{\partial z}\right) + \delta e, \tag{75}$$

where $\delta c$, $\delta e$ are the perturbations of the concentration, the emission rate of a species respectively. For vertical diffusion, $K(z)$ is a differentiable function of height $z$.

For velocity $v_x = v_y = 0.5$ and the time step $\triangle t = 0.5$, the numerical solution is based on the symmetric operator splitting technique (Yanenko, 1971) with the following operator sequence

$$\delta c(t + \triangle t) = T_x T_y D_z A D_z T_y T_x \delta c(t), \tag{76}$$

where $T_x$ and $T_y$ are transport operators in horizontal directions $x$ and $y$, $D_z$ is the diffusion operator in vertical direction $z$. The parameters of emission and deposition rates are included in $A$. The Lax-Wendroff algorithm is chosen as the discretization method for horizontal advection with $\triangle x = \triangle y = 1$. The vertical diffusion is discretized with $\triangle z = 1$ by Crank-Nicolson scheme with the Thomas algorithm (Higham, 2002) as solver. The number of the grid points is $N_g = 1125$.

With the same temporal and spacial discretization of the concentration, the background knowledge of the emission rate is given by $e_b(t_n, i, j, l)$, where $n = 1, \cdots, N$. We establish the discrete dynamic model of the emission rate according to (8)

$$\delta e(t_{n+1}) = M_e(t_{n+1}, t_n)\delta e(t_n), \quad n = 1, \cdots, N, \tag{77}$$

where $M_e(t_{n+1}, t_n) = e_b(t_{n+1})/e_b(t_n)$.





For expository reasons we assume $\delta d$ is a constant over time and the only one fixed observation configuration is time-invariant. It indicates that the observation operator mapping the state space to the observation space is a $1 \times 2N_g$ time-invariant matrix.

In our simulations, we produce $q = 500$ (the ensemble size number) samplings for the initial concentration and emission rate

respectively by pseudo independent random numbers and make the states correlated by the moving average technique. In the following, we present three different tests, aiming to demonstrate the roles of variable winds, emissions, and vertical diffusion.

*Advection test:* The following part demonstrates the application of the DFS analysis tool by basic examples, designed to show the expected elementary outcomes of the following situations, which exhibit the effects of assimilation window length in relation to emission location: these include (i) an assimilation window too short to capture emission impacts at the observation

site, (ii) an extended assimilation window with balanced signal of impacts of concentrations and emissions at the observation site, and (iii) a further increased assimilation window featuring a declining impact of initial values in favour of growing emission impact. The first elementary advection test (Fig. 1 to Fig. 7) identifies the sensitivities of parameters subject to different wind direction and data assimilation window (DAW) through the DFS. Focusing on the advection effects, we apply the model with a weak diffusion process ($K(z) = 0.5e^{-z^2}$).

In Fig. 1 to 3 we assume southwesterly winds and the assigned data assimilation windows are $10\triangle t$, $35\triangle t$ and $48\triangle t$ respectively. The contributions to EnDFS of the initial states are shown in the left panels of Fig. 1 to 3. We can find that the horizontal fields at lowest layer ($z = 0$) where the estimates of the concentration are probably improved is enlarged with the extension of data assimilation windows. It is because that more and more grid points of the concentration are correlated with longer data assimilation windows.

The right panels of Fig. 1 to 3 show the EnDFS of the emission rate at each grid point with $z = 0$. From Fig. 1, we can observe that the EnDFS of the emission rate are smaller than the case of initial value in the influenced area. It indicates that the observations cannot detect the emission rate within $10\triangle t$ data assimilation window. Thus, in this case initial values alone can be optimized. It is shown in the right panels of Fig. 2 and Fig. 3 that the emission rate plays a more and more important role on the impact of observations. In this two cases, we consider both the concentration and emission rate as optimized parameters.

The quantitative balance between the concentration and emission rate is provided in Table 1.

Fig. 4 exhibits in its upper row panels the singular values underlying the results shown in Fig. 1 to 3. We approximate the sensitivities of the initial concentrations by the first five leading singular vectors weighted by the associated singular values in the nuclear norm and show the results in the three panels of Fig. 4, lower row. It is clearly visible that the sensitive area can be well targeted by only few singular vectors, although the sensitivity analysis cannot provide the quantitative solutions with

a clear statistical significance as the the degree of freedom for signal of the model. The areas of influence to the measurement site in depencence of wind direction and assimilation window lengths is clearly visualized, corresponding to expectations.

As counter example, Fig. 5 to 7 also show the EnDFS of the concentration and emission rate under the same assumptions as Fig. 1 to 3 respectively, except that northeasterly wind is assumed. Clearly, with the northeasterly wind, whatever the duration of the assimilation window is, the emission is not detectable and improvable by that particular observation configuration. This

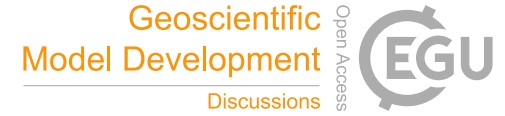

hypothesis is demonstrated by our method. The quantitative balances are exposed in Table 1 for the related figures, where the insensitivity to emission rate optimisation remains equally low and induced by numerical noise. .

*Emission signal test:* The purpose of emission signal test (Fig. 8 and Fig. 9) is to assess the impact of observation configurations to the emission rates evolved with different diurnal profiles. We have the same assumptions as Fig. 3 except the wind speed in Fig. 8 and Fig. 9 is accelerated such that the profiles of the emission rate is better detectable as to the observation. The only distinction between the situations in Fig. 8 and Fig. 9 is the pronounced diurnal cycle background profile of the emission rate during the assimilation window $48\triangle t$, schematically simulating a rush hour induced source. Since the profiles of emission rates are correlated with the emitted amount of that species during the data assimilation window, it is clearly shown in Table 1 that the distinct variation of the emission rate during the data assimilation window acts to level $\bar{p}^c$ and $\bar{p}^e$, and thus helps to improve the estimates of source.

*Diffusion test:* The diffusion test (Fig. 10 to Fig. 12) aims to test the approach via comparing the EnDFS of the concentration and the emission rate at the layer $z = 0$ with a weak diffusion process and a strong diffusion process. We assume the observation configuration at each time step is located at $(12, 10, 4)$ in the diffusion test , with $K(z) = 0.5e^{-z^2}$ in Fig. 10 and $K(z) = 0.5e^{-z^2} + 1$ in Fig. 11. Besides, Fig. 10 and Fig. 12 preserve the same assumptions with Fig. 3.

It is obviously seen from Fig. 3 and Fig. 10 that the different observation locations strongly influence on the distribution of the concentration. Table 2 shows that with the same diffusion coefficient the degree ensemble degree of freedom for signal of the concentration in the lowest layer in Fig. 3 is definitely larger than the one for Fig. 10. Moreover, it can be seen from Table 1 that the observation configuration at the top layer is not efficient to the emission rate with such weak diffusion within $48\triangle t$ data assimilation window .

Comparing Fig. 10 with Fig. 11, we can find that the EnDFS of concentration and emission rate increase with the stronger diffusion process. The increasing impact of the observation configuration with the stronger diffusion is also verified by the EnDFS and ensemble relative ratios of DFS of the concentration and emission rate for Fig. 10 and 11 in Table 2. The balances between the concentration and emission rate for Fig. 10 and 11 are shown in Table 1. The significant difference of the 'weight' of emission rate in Table 1 implies that the observation configuration cannot detect the emission at the lowest layer with such a weak diffusion in Fig. 10 and with the stronger in Fig. 11 both the concentration and emission rate should be considered as optimized parameters with the corresponding 'weights'.

Finally, similar to Fig. 4, the singular values of Fig. 10, Fig. 11 and the approximating targeting results of sensitive parameters are shown in Fig. 12.

## 8 Conclusions and Outlooks

It is novel in this study that the tangent linear form of the atmospheric transport model was extended by emission rates under the assumption that emissions preserve the invariant diurnal profiles. The homogeneity of the extended model provides the initial value and emission rates the equivalent statuses such that emission rates are available to be optimized by Kalman smoother as model variables. In the context of Kalman smoother, the contribution to the degree of freedom for signal is derived as




the criterion to evaluate the potential improvement of each grid point in the state vector and calculated by singular value decomposition. With a statistical interpretation, we can apply it to determine in advance, which parameters can be optimized by the data assimilation procedure. The degree of freedom for signal and a number of metrics provide us with the quantitative solutions to measure to what extend the parameters can be optimized. Due to its relativity, it is uniformly available for any

prior initial values of invertible background covariances. Further, the proposal of the ensemble method, based on EnKS, gives us a computationally feasible access to assess the degree of freedom for signal.

The sensitivity of observational networks was formulated by seeking the fastest directions of the perturbation ratio between initial states and observation configurations during the entire time interval. An elementary advection-diffusion example illustrated the significance of relative improvements covariances and their various metrics in different situations and compared them

with the results of the sensitivity analysis.

In the future, on one hand, we plan to apply the efficiency analysis into the real atmospheric transport model to solve practical problems. On the other hand, we will get deeper insight to the sensitivity analysis for wider applications. For example, in order to evaluate the impact of observations in some certain locations, the local projection operator introduced by Buizza and Montani (1999) can be applied into approaches in Section 5 and Section 6.

## 9   Code availability

The code can be obtained by contacting the corresponding author.

*Acknowledgements.*   This work was supported by HITEC and IEK-8 of Forschungszentrum Jülich.



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





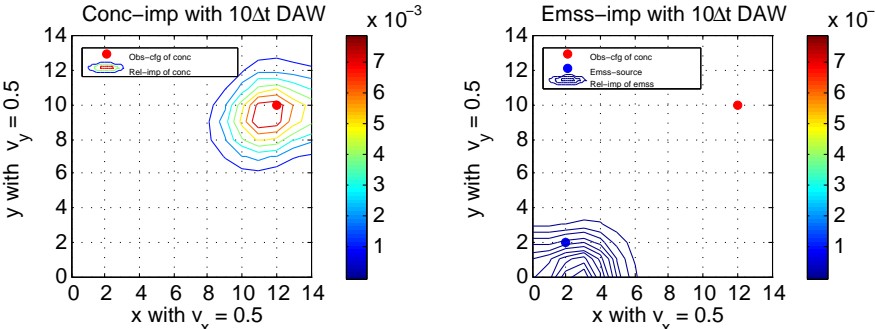

**Figure 1.** Advection test with $10\triangle t$ DAW and southwesterly wind. Isopleths of ensemble relative improvements of the concentration and emission rate are shown in the left and right figure panels respectively. The point located at $(12,10,0)$ named as 'Obs-cfg of conc' shows the invariant observation configuration. The point located at $(2,2,0)$ named as 'Emss-source' is the source of the emission rate.

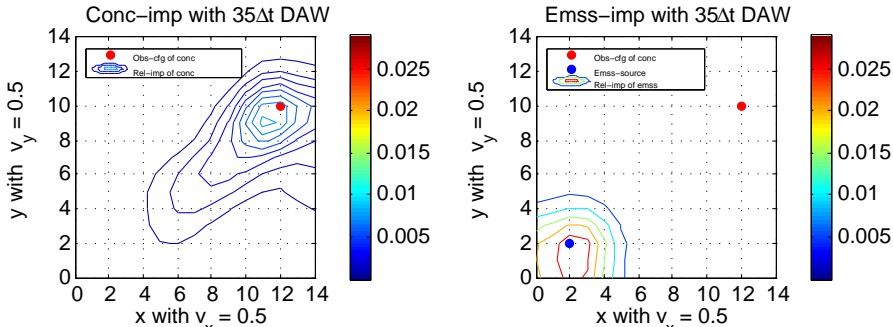

**Figure 2.** Advection test with $35\triangle t$ DAW and southwesterly wind. Plotting conventions are as in Fig. 1.

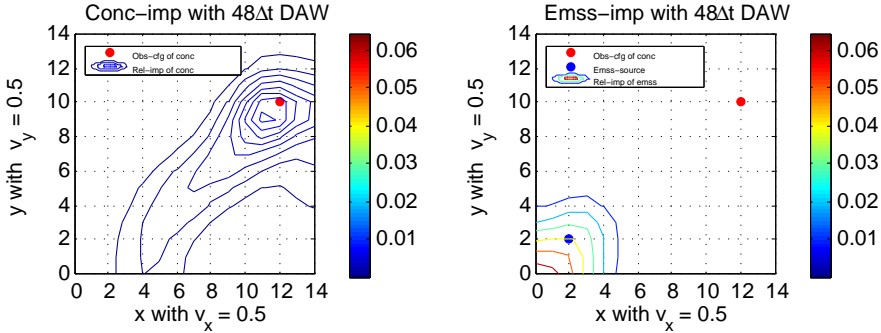

**Figure 3.** Advection test with $48\triangle t$ DAW and southwesterly wind. Plotting conventions are as in Fig. 1.



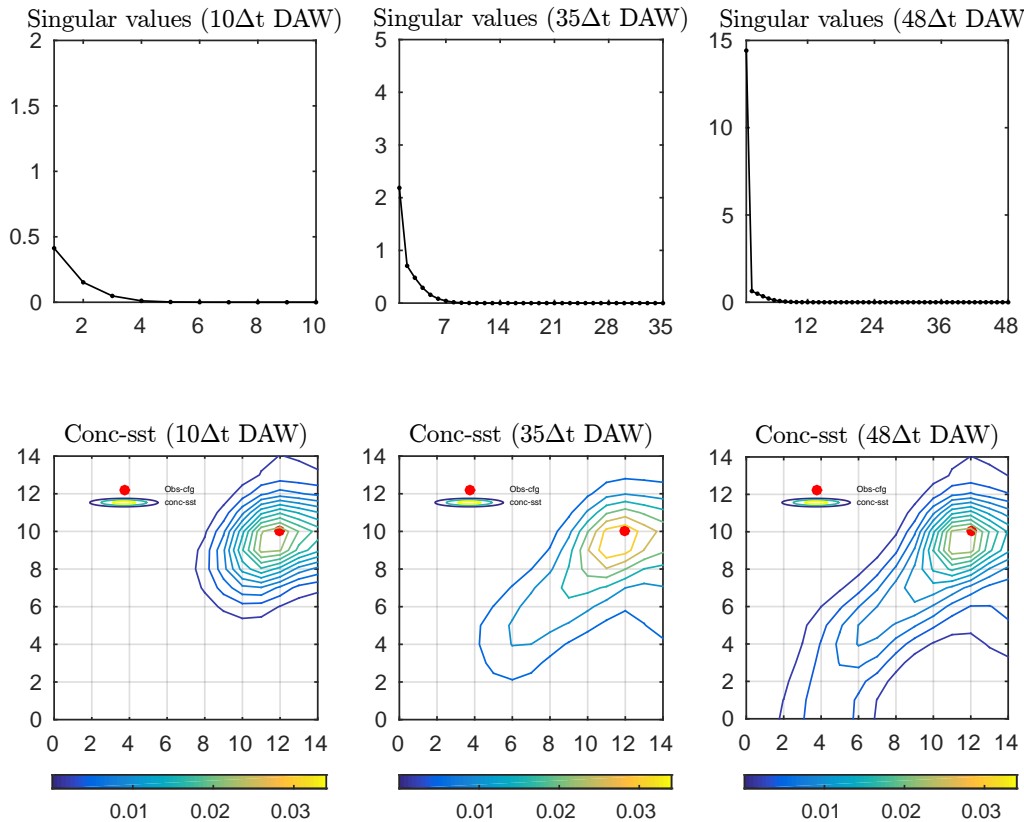

**Figure 4.** Singular values of Fig. 1 ∼ 3 and sensitivities of initial states approximated by 5 leading singular values.

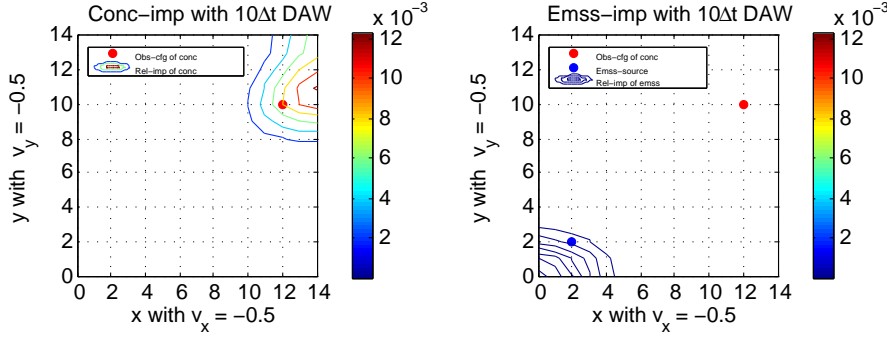

**Figure 5.** Advection test with $10\triangle t$ DAW and northeasterly wind. Plotting conventions are as in Fig. 1.





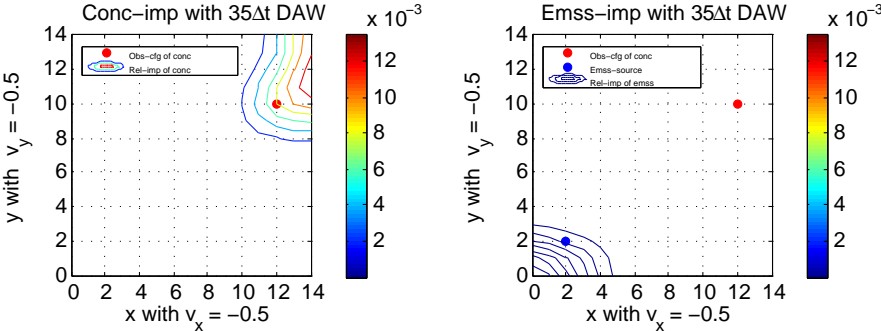

**Figure 6.** Advection test with $35\triangle t$ DAW and northeasterly wind. Plotting conventions are as in Fig. 1.

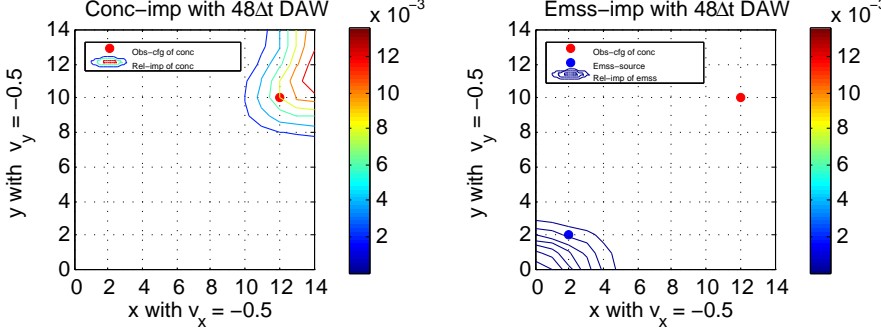

**Figure 7.** Advection test with $48\triangle t$ DAW and northeasterly wind. Plotting conventions are as in Fig. 1.

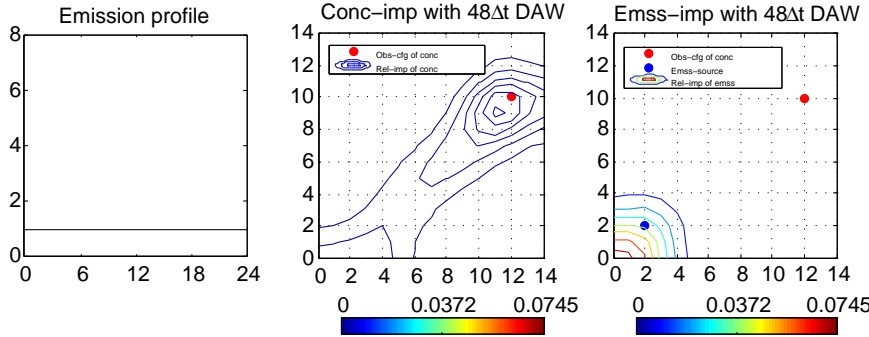

**Figure 8.** Emission signal test (weak) with $35\triangle t$ DAW and southwesterly wind ($v_x = 1$ and $v_y = 1$). Plotting conventions are as in Fig. 1.

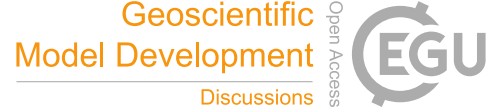



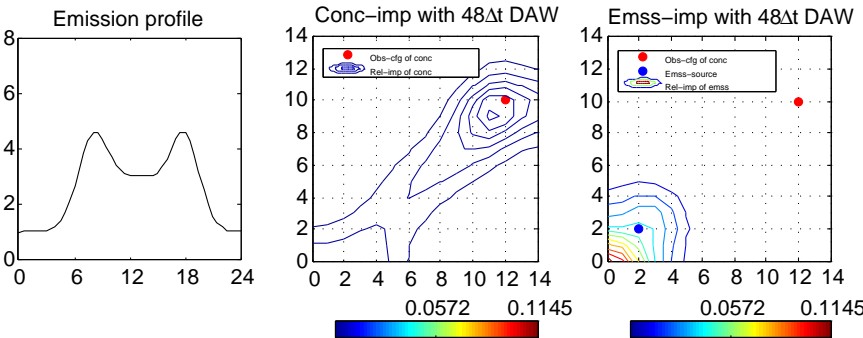

**Figure 9.** Emission signal test (strong) with $48\triangle t$ DAW and southwest wind ($v_x = 1$ and $v_y = 1$). Plotting conventions are as in Fig. 1.

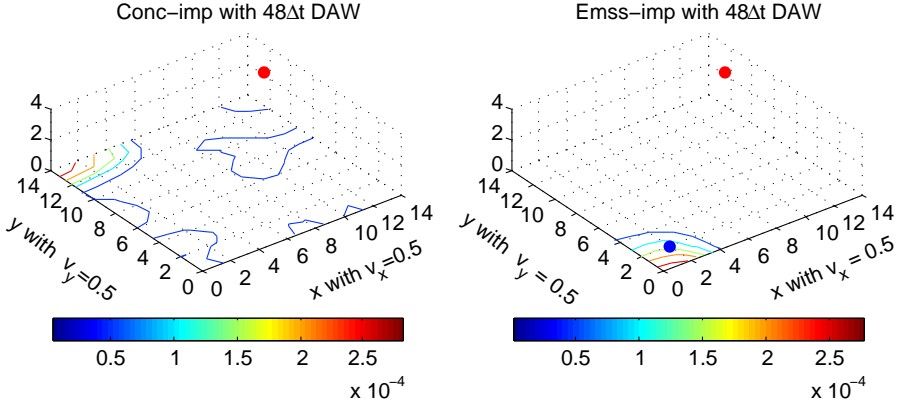

**Figure 10.** Diffusion test (weak) with $48\triangle t$ DAW and southwesterly wind. Plotting conventions are as in Fig. 1.

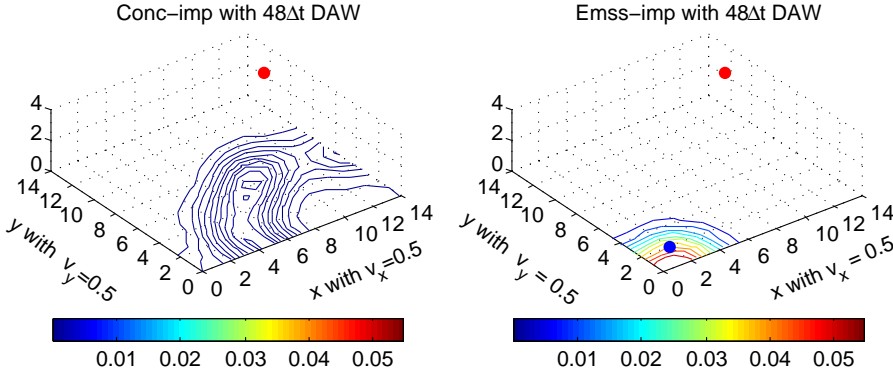

**Figure 11.** Diffusion test (strong) with $48\triangle t$ DAW and southwesterly wind. Plotting conventions are as in Fig. 1.





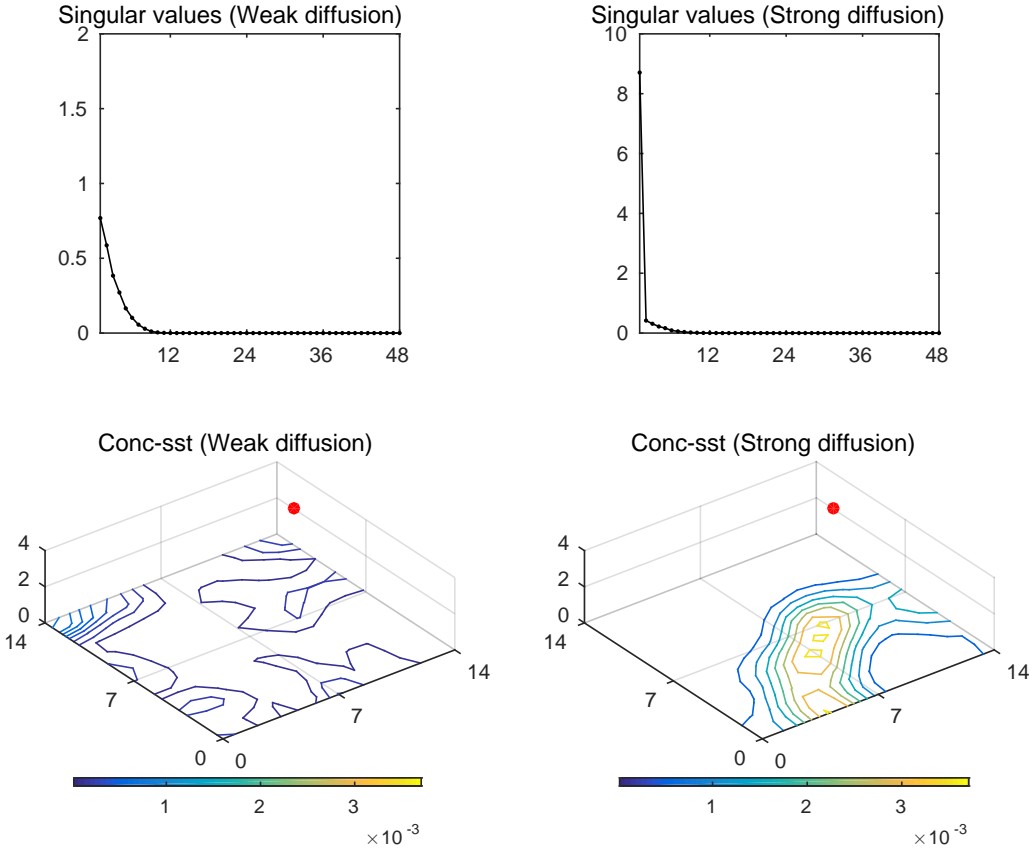

**Figure 12.** Singular values of Fig. 10 and Fig. 11 and sensitivities of initial states approximated by 5 leading singular values.

**Table 1.** Ensemble relative ratios of the initial value and emission rate at the lowest layer.

|  | Fig. 1 | Fig. 2 | Fig. 3 | Fig. 5 | Fig. 6 | Fig. 7 | Fig. 8 | Fig. 9 | Fig. 10 | Fig. 11 |
|---|---|---|---|---|---|---|---|---|---|---|
| $\bar{p}^c$ | 0.9920 | 0.4548 | 0.2848 | 0.9937 | 0.9939 | 0.9939 | 0.2248 | 0.1808 | 0.9928 | 0.1905 |
| $\bar{p}^e$ | 0.0080 | 0.5452 | 0.7152 | 0.0063 | 0.0061 | 0.0061 | 0.7752 | 0.8192 | 0.0072 | 0.8095 |

**Table 2.** The ensemble degrees of freedom for signal of the initial value and emission rate at the lowest layer.

|  | $\bar{P}^c_{low}$ | $\bar{P}^e_{low}$ |
|---|---|---|
| Fig. 3 | 0.2767 | 0.8851 |
| Fig. 10 | 0.0102 | 0.0030 |
| Fig. 11 | 0.0500 | 0.7892 |