# Peer review of "The degree of freedom for signal assessment of measurement networks for joint chemical state and emission analysis"

_Geoscientific Model Development, 2017_

## Short Comment (SC1) · 10 Oct 2017

Xueran

As explained in https://www.geoscientific-model-development.net/about/manuscript_types.html GMD is expecting that authors upload the program code of models as a supplement or make the code available at a data repository preferable with an associated DOI (digital object identifier) for the exact model version described in the paper. If for some reason your code cannot make available in this form as the code availability section suggests you need to state the reasons why the source code is not available.

It is also not clear in which environment your implementation is implemented and what is required to run the code. Could you please clarify?

All the best Lutz Gross GMD Executive Editor

---

## Short Comment (SC2) · 23 Oct 2017

Xueran

Sorry, I have just became aware that the code is provided as supplement. However, the paper still suggests to contact the author. Please fix this in the next version of the paper.

Lutz Gross GMD Executive Editor

---

## Author Comment (AC1) · 23 Oct 2017

Dear Prof. Gross,

thanks for your comments. We will correct it in the next version.

Best regards, Xueran

---

## Referee Comment (RC1) · Anonymous Referee #1 · 28 Feb 2018

General comments: This paper describes a methodology to determine the information content (in particular, the degree of freedom for signal) of joint chemical state and emission inversions. A detailed mathematical analysis is provided in the first part, followed by numerical illustrations. There are several issues with this article: 1) The method is known. Recent articles, such as, e.g., Bousserez and Henze (qjrms, 2017), Spantini et al. (2015), presented in detail this approach for information content analysis, with similar mathematical developments. Other related methods in the context of ensemble data assimilation are described in the literature, see for instance Anderson (2001) (Ensemble Adjustement Kalman Filter). This has to be acknowledged and discussed by the authors. 2) The grammar needs to be thoroughly checked. In many places the text
is unclear due to poor phrasing. 3) The main text contains too many equations, which is distracting and makes the reading difficult. Most of the mathematical developments should be moved to an Appendix.

Detailed comments: 1) Introduction: It is too long. Also, there are lots of redundancies. A number of references should be added (see general comments above for some of them) and discussed. In particular, the authors should clearly acknowledge previous works where similar analysis were conducted, and explain what their study adds to the current state of knowledge. If there is no real novelty in the approach, then the article should be presented as a review paper focusing on methods for information content analysis, with a numerical application for the specific problem of joint chemical state and emission inversion. 2) P1, L17-18: Rephrase. 3) P1, L19-24: Shorten. There are many repetitions. 4) P6, L13-23: Could be simplified (or should go to an Appendix). 5) P7 L9-15: Redundancies. Poor phrasing. 6) P7 L19: Good. You should do that simplification earlier. There is no need to split all the operators like in (11). 7) P7, Eq (17): Please define the mathematical terms you use. For instance, t {-1} is not defined here. Presumably  $P^{-1}(t \ 0|t \ -1)$  is the prior error covariance matrix, in which case it should be clearly stated. 8) P8, L1-2: I do not understand this sentence. 9) P8. Eq (19): this equation is well-known and the previous developments are not needed. 10) P8 L13-14: Unclear. Please rephrase. 11) P13, Eq (53): Notation V, S, U has already been used in (46) and (48), and the SVDs in (53) and (46) are not related. Please use another notation. 12) Section 6: Again, lots of mathematical developments that should be in an Appendix.

---

## Author Comment (AC2) · 3 Jul 2018

Dear Reviewer,

Thank you for your attention to our manuscript, and your valuable comments on the research, as well as the suggestions for improving the paper. We have tried to address your concerns. Details on the changes are below.

General comments

*- This paper describes a methodology to determine the information content (in particular, the degree of freedom for signal) of joint chemical state and emission inversions.*

[Figure]

*A detailed mathematical analysis is provided in the first part, followed by numerical illustrations. There are several issues with this article: 1) The method is known. Recent articles, such as, e.g., Bousserez and Henze (qjrms, 2017), Spantini et al. (2015), presented in detail this approach for information content analysis, with similar mathematical developments. Other related methods in the context of ensemble data assimilation are described in the literature, see for instance Anderson (2001) (En- semble Adjustement Kalman Filter). This has to be acknowledged and discussed by the authors.*

**Thanks for reminding us about those references. We will definitely cite them in the following version of our paper.**

**There are indeed several similar equations. For example, we can find the same equation, such as Eq.(17) and (19), in our paper and the references by Spantini et al. (2015) and Bousserez and Henze (2017), which are the expression of the inverse of posterior covariance matrix. But those equations are well known and obtained from several previous papers, such as paper from Li and Navon, 2001 we cited. Besides, we use the singular value decomposition into the same matrix as the paper by Bousserez and Henze (2017). However, there are some points need to be emphasized about the similarity to the two references, differences and novel points of our paper.**

- **As to the similarity, we firstly have to clarify that the previous version of this paper has been published online in arXiv.org as a preprint with the title 'Efficiency and Sensitivity Analysis of Observation Networks for Atmospheric Inverse Modelling with Emissions' on Mar. 23, 2015. Then we reorganized the some part of the text and terminology in order to improve the preprint and got the current version submitted to GMD. Thus, the few same expression of the posterior covariance matrix as the one in paper by Spantini et al. (2015) is a coincidence. Besides, the similar equations from the paper by Bousserez and Henze (qjrms, 2017) are actually obtained by us earlier**

[Figure]

(the preprint on Mar. 2015).

- Spantini et al. aimed at to approximate the posterior covariance in a sub-space. It differs from our motivation to detect the capacity of a given observation network to optimize the state vector and emissions with the help of posterior covariance. It is worth noting that it is novel in our paper that we extended the state vector of the linearized atmospheric transport model so that the emissions were included in the model-state vector and the model turned to be homogeneous (Section 2). In section 3 and 4, based on the above extended model, our idea is to consider the normalization of the difference between the background forecast error covariance matrix and the analysis error covariance matrix as a criteria to investigate the capacity of observation networks to improve the estimation of both the concentrations and emissions. The normalization is crucial because it provides an uniform standard to any cases with diverse initial conditions and allows us to apply our approach into them directly and not considered in the paper by Spantini et al.. To be specific, compared with Eq. (3.6), (3.10), (4.4) and (4.13) in Spantini et al. (2015), the similar-looking equations (74), (72), the equation in line 13 on Page 9 and Eq. (54) in our paper are distinct.

- Bousserez and Henze discussed the theoretically equivalent approach to Bocquet et al (2011) they cited, yet different interpretation in terms of projection and aggregation framework. Both approaches, which can be considered, with some limits, as dual to each other, seek to optimize, in a controlled way, the complexity of the underlying problem space. The observational network is only involved by controlling the error of representatives. In contrast, our approach seeks to quantify the degree of freedom for signals for a given observation network and model/analysis grid with respect to its value for a heterogeneous parameter optimization (emission strengths and initial values in our case). In some sense, this can be considered as a test

**for prolongation of observational data to an extended parameter optimization space.**

- **Besides, in section 5, we originally derived the ensemble case of the approach in the section 3 and 4 without the non-singularity of the background covariance matrix, which differs from the derivation in the paper by Anderson (2001).**

*- 2) The grammar needs to be thoroughly checked. In many places the text is unclear due to poor phrasing.*

**Thanks for your comments. We will go through the whole paper to correct the grammar mistakes and improve the phrasing and attempt to provide a satisfactory writing in the next version.**

*- 3) The main text contains too many equations, which is distracting and makes the reading difficult. Most of the mathematical developments should be moved to an Appendix.*

**Thanks for your valuable advise. Most of the mathematical developments will be moved to an Appendix and then the text will be reorganized.**

*Detailed comments:*
*- 1) Introduction: It is too long. Also, there are lots of redundancies. A number of references should be added (see general comments above for some of them) and discussed. In particular, the authors should clearly acknowledge previous works where similar analysis were conducted, and explain what their study adds to the current state of knowledge. If there is no real novelty in the approach, then the article should be*

*presented as a review paper focusing on methods for information content analysis, with a numerical application for the specific problem of joint chemical state and emission inversion.*

**Thanks for the comments. The introduction is shortened and the novelties are emphasized.**

*-2) P1, L17-18: Rephrase.*

**We changed this to: Parameter mis-specfications in a model can only be identified within data assimilation intervals of space-time methods, if the the simulation is sufficiently sensitive and the error related observability of the mesurement network is given.**

*-3) P1, L19-24: Shorten. There are many repetitions.*

**These sentences are now replaced by : "Otherwise, the forecast degrades beyond the observation controlled period.**

*-4) P6, L13-23: Could be simplified (or should go to an Appendix).*

**P6, L13-23 has been simplified and P7, L1-L7 has been put into Appendix A.**

*-5) P7 L9-15: Redundancies. Poor phrasing.*

**All sentences except the last are removed.**

*-6) P7 L19: Good. You should do that simplification earlier. There is no need to split all*

*the operators like in (11).*

**Thanks for your comments. Though Eq. (15) and (16) are the general case of Eq. (11), Eq. (11) is novel in our paper.**

*-7) P7, Eq(17): Please define the mathematical terms you use. For instance, $t_{-1}$ is not defined here. Presumably $P^{-1}(t_0|t_{-1})$ is the prior error covariance matrix, in which case it should be clearly stated.*

**The required definitions are inserted.**

*-8) P8, L1-2: I do not understand this sentence.*

**It has been deleted.**

*-9) P8. Eq(19): this equation is well-known and the previous developments are not needed.*

**We arrive at a more comfortable notation.**

*-10) P8 L13-14: Unclear. Please rephrase.*

**We changed it into: "We aspire an expression, which allows for a direct and normalized comparison of sensitivities to initial values and emission rates. "**

*-11) P13, Eq (53): Notation V, S, U has already been used in (46) and (48), and the SVDs in (53) and (46) are not related. Please use another notation.*

[Figure]

**The notation has been changed accordingly.**

*-12) Section 6: Again, lots of mathematical developments that should be in an Appendix.*

**Most mathematical developments have been moved into Appendix B.**

Sincerely,

Xueran Wu (corresponding author), Hendrik Elbern and Birgit Jacob

---

## Referee Comment (RC2) · Anonymous Referee #2 · 22 Aug 2018

This paper discusses an approach to calculate the degrees of freedom for signal given by observation networks in joint state-emission data assimilation. The problem is important, and therefore deserves a careful treatment. The paper is not particularly well written, and I would like to encourage the authors to revise the language and the spelling. Also, the presentation is somewhat repetitive and can be shortened for an easier read. While the title says "chemical", there is no discussion of nonlinear dynamics in either the theory or the numerical experiments.

Major comments: 1. Please explain in detail how the methodology proposed in Section 5 is different than the following work (if it is not, explain similarities and

cite appropriately): https://rmets.onlinelibrary.wiley.com/doi/pdf/10.1002/qj.123
https://www.sciencedirect.com/science/article/pii/S1877050912002347
http://www.met.rdg.ac.uk/phdtheses/Information%20content%20of%20observations%20in%20variational%20data%20ass

2. Please explain the computational cost of the methodology. How does the cost scale with the number of ensembles? With the data set? With the model size? With the assimilation window length?

3. There is no comparison between the results obtained with the authors' approach and other existing approaches in the literature, e.g., Zupanski 2007. The numerical experiments would be more convincing if such a comparison was included.

4. There is no discussion on how this can be applied to nonlinear systems. Adding some nonlinear chemistry to one of the test problems may help argue that this methodology, while developed under linear assumptions, can be in fact useful for nonlinear systems as well.

Minor comments:

Eqn. (25) does not seem to be the traditional 1-norm of a matrix. Please clarify the notation. Also, clarify what matrix square root is used, as there are infinitely many possibilities. Eqn. (38), for example, does not seem to follow from the current (25) unless we are more specific.

Equation (20), /*we* define a matrix P.../ This matrix is the standard starting point in the definition of DFS, cf. Fisher 2003, Singh 2013. The discussion of Eqn (28) is confusing. The well accepted meaning of Ptilde(j) in the literature is: how much have we learned about variable xj from the data: from 0 (nothing) to 1 (everything). This is the amount learned about one degree of freedom (xj) out of n; the total number of degrees of freedom informed by the data/signal is the sum over all variables.

Editorial comments: Please carefully revise the writing of the manuscript (English correctness) as well as the spelling. There are hard to read (in English) formulations such

as /aspiring a means/ (?). There are also typos in the manuscript. For example: /to what extend/ should be /to what extent/ in the Abstract, or /anfd Wu/ should be /and Wu/ in the Introduction, etc. Please avoid embedding URLs in text, they are best deferred as citations referring to web pages.

―――――――――――――――――

---

## Author Comment (AC3) · 30 Aug 2018

Dear Reviewer,

Thank you for your attention to our manuscript, and your valuable comments on the research, as well as the suggestions for improving the paper. We have tried to address your concerns. Details on the changes are below.

*Major comments:*

*- 1. Please explain in detail how the methodology proposed in Section 5 is different than the following work (if it is not, explain similarities and cite appropriately):*

[Figure]

*(1)https:// rmets.onlinelibrary.wiley.com/ doi/ pdf/ 10.1002/ qj.123*
*(2) https:// www.sciencedirect.com/ science/ article/ pii/ S1877050912002347*
*(3)http:// http:// www.met.rdg.ac.uk/ phdtheses/ Information%20content%20of%*
*20observations%20in%20variational%20data%20assimilation.pdf*

**Thanks for reminding us about those references. We will definitely cite them in the following version of our paper.**

**The central difference of our study to those cited is the problem of model control by two (and generalizable to multiple) markedly different parameters: here initial values vs. emission rates (boundary condition) for chemistry transport models. The resulting outcome is expressed as partial DFS, related to each parameter, and dependent on the measurement configuration and transport (weather conditions). The proposed algorithm quantifies each sensitivity. More specific differences to the papers to be considered include:**

**The paper with the link (1) studied an ensemble-based data assimilation approach for Kalman filter and 3D-VAR. The studies of our paper are based on Kalman smoother and 4D-VAR.**

**The paper by Singh et al. (2013) we cited is actually a more complete version of the paper with the link (2). They used the information metrics to qualify the value of measurements in frame of ensemble runs based on 4D-VAR. However, in section 5 of our paper, all derivations are based on an ensemble Kalman smoother, which are different from 4D-VAR for nonlinear models. Besides, we are not only interested in the DFS of the total system, but also focus on improvement of each grid point from any given observation network and give metrics for concentrations and emissions separately. Further, we apply the singular value decomposition in Eq (53) to calculate $\tilde{P}$ without the calculation of Hessian of the cost function based on 4D-VAR.**

**In the thesis with link (3), the author studied the information content of obser-**

**vation based on 3D-VAR and 4D-VAR. But it is hard to see the ensemble-based studies in that thesis. Besides, the author of the thesis used the SVD technique into the observability matrix, which is different from Eqn (50) in our paper.**

*- 2. Please explain the computational cost of the methodology. How does the cost scale with the number of ensembles? With the data set? With the model size? With the assimilation window length?*

**Thanks for your comments. It shows in our tests that the computational cost is linearly increasing with the increasing of ensemble number and the assimilation window length. Besides, since the dimensions of the data set and model size are much larger then the ensemble numbers, the computation cost of our approach related to SVD (Eqn. 49) mainly depends on the numbers of ensembles. The explanation about the computational cost has been added in the example with the content " It has been tested that the computation cost of our approach is linearly increasing with the number of ensembles.(P14, line 16) " and "The computation times are approximately 8.1s, 28.5s and 39.4s in our tests with the above three different assimilation windows, from which we can verify that the computation cost is nearly linearly increasing with the length of data assimilation window. (P14, line 28)"**

*- 3. There is no comparison between the results obtained with the authors' approach and other existing approaches in the literature, e.g., Zupanski 2007. The numerical experiments would be more convincing if such a comparison was included.*

**Thanks for your suggestion. We didn't compare the results with other works just because it is novel to extend the transport model with emissions in our paper and we more focus on the quantitative balance between the original model state**

and emissions. At the same time, we completely agree that it would be more convincing if we would compare our results with the others, once studies with a similar objective are available. We believe that it will be more significant to do the comparison if the approach is applied into the real atmospheric models. In fact, this is exactly what we are doing now (P21, line 16).

*- 4. There is no discussion on how this can be applied to nonlinear systems. Adding some nonlinear chemistry to one of the test problems may help argue that this methodology, while developed under linear assumptions, can be in fact useful for nonlinear systems as well.*

Thanks for your valuable advise. We mainly focus on the linear model and only give a linear example in this paper. We are so sorry that we didn't clearly explain that this approach can be also applied into nonlinear case. In general, the scope and limits of our approach is exactly the same as 4D-var and Kalman(-Bucy)-filters apply for non-linear (chemical) models. The central assumption is the validity of the linearized (tangent-linear) model. In the context of atmospheric chemistry modelling, the simulation must "be in the right chemical regime". (For a detailed discussion see papers Goris and Elbern, 2013 and 2015.) However, for critical cases of photochemistry with, say, observations of ozone as non-emitted product species, driven by emitted nitrogen oxides and volatile organic compounds, it can be safely expected that still the measurement network and transport conditions invoke the same results as in case of a passive tracer. A short discussion is given on Page 12 , 13 with Eqn (56) in the revised manuscript.

*Minor comments:*
*Eqn. (25) does not seem to be the traditional 1-norm of a matrix. Please clarify the notation. Also, clarify what matrix square root is used, as there are infinitely many*

*possibilities. Eqn. (38), for example, does not seem to follow from the current (25) unless we are more specific.*

**Thanks for your reminding and sorry for the unclear notation. This is the nuclear norm of a matrix. Avoiding the confusion with the induced matrix 1-norms, we directly called it as the nuclear norm in the paper (Please see Page 7, line 30 in the revised manuscript). Besides, the definition of matrix square root is clarified in the revised version (Please see Page 6, line 25).**

*Equation (20), /\*we\* define a matrix P: : :/ This matrix is the standard starting point in the definition of DFS, cf. Fisher 2003, Singh 2013. The discussion of Eqn (28) is confusing. The well accepted meaning of Ptilde(j) in the literature is: how much have we learned about variable xj from the data: from 0 (nothing) to 1 (everything). This is the amount learned about one degree of freedom (xj) out of n; the total number of degrees of freedom informed by the data/signal is the sum over all variables.*

**Thanks for the comments and sorry for unclear expression. We have rewritten this part. Please see Page 6, from line 21 to line 2 on Page 7.**

*Editorial comments: Please carefully revise the writing of the manuscript (English correctness) as well as the spelling. There are hard to read (in English) formulations such as /aspiring a means/ (?). There are also typos in the manuscript. For example: /to what extend/ should be /to what extent/ in the Abstract, or /anfdWu/ should be /andWu/ in the Introduction, etc. Please avoid embedding URLs in text, they are best deferred as citations referring to web pages.*

**Thanks for the comments. We went through the whole paper to correct the grammar mistakes and improve the phrasing and attempt to provide a satisfactory writing in the next version. Due to the shortness of introduction, the URLs**

**have been deleted.**

Sincerely,

Xueran Wu (corresponding author), Hendrik Elbern and Birgit Jacob

---

## Author Comment (AC4) · 30 Aug 2018

**The degree of freedom for signal assessment of measurement networks for joint chemical state and emission analysis**

Xueran Wu[1,2,3], Hendrik Elbern[1,2], and Birgit Jacob[3]

[1]IEK-8, Forschungszentrum Jülich, Wilhelm-Johnen-Straße, 52428, Jülich, Germany.
[2]Rhenish Institute for Environmental Research at University of Cologne, Aachener Straße 209, 50931, Cologne, Germany.
[3]Mathematics Department, Unversity of Wuppertal, Gaußstraße 20, 42119, Wuppertal, Germany.

*Correspondence to:* Xueran Wu (x.wu@fz-juelich.de)

**Abstract.** The degree of freedom for signal (DFS) is generalized and applied to estimate the potential observability of observation networks for augmented model state and parameter optimisation problems. Control parameters may include prognostic state variables, mostly the initial values and insufficiently known model parameters. As for chemistry-transport models, emission rates are at least as important as initial values for model evolution control. Extending the optimization parameter set must be met by appropriate observation networks, which only allow for controlling the full optimization task, if the observability is satisfied. In this paper,we introduce an approach to quantify the degrees of freedom for signal jointly but individually for initial trace gas state and emission rates for transport-diffusion models forced by emissions. By applying a Kalman smoother, the quantitative assessment method on the efficiency of observation configurations to optimize the initial value and emission rates is developed based on the singular value decomposition. For practical applications, we derive an ensemble based version of the approach and give several elementary experiments.

**1   Introduction**

Air quality and climate change are influenced by the fluxes of green house gases, reactive gas emissions and aerosols. The temporal evolution of chemistry in the atmosphere is usually modelled by atmospheric chemistry transport models. poorly known initial values, sources and sinks are a serious problem for the quality of simulation, which can be addressed by data assimilation and inverse modelling. Parameter mis-specfications in a model can only be identified within data assimilation intervals of space-time methods, if the simulation is sufficiently sensitive and the error related observability of the measurement network is given. This poses the observability problem. Otherwise, the forecast degrades beyond the observation controlled period.

In practice data assimilation problems are typically solved in circumstances, where the number of observations are markedly lower than the model degree of freedom (Daley 1991). Consequently, when aiming to improve the quality of analysis by observation configurations, several aspects can be considered. These include (i) to optimize the observation network, subject to given constraints, (ii) to evaluate the value of individual or types of observations for the analyses, and (iii) to quantify the degree of which the analysis can be influenced by the observations, which is related to the sensitivity.

The observation network optimization problem (i) has been addressed traditionally by Observation System Simulation Experiments (OSSEs, e.g Daley 1991). The advanced concept of targeted observations has been popularized with the FASTEX campaign (e.g Szunyogh et al. 1999; Langland et al. 1999). Theoretical studies are presented, for example by Bishop et al. (1999); Berliner et al. (1998), or recently by Bellsky et al. (2014) for highly nonlinear dynamics, and Wu et al. (2016) for the

5  optimal locations of observations for time-varying system within a finite-time interval.

The benefit assessment (ii) of individual observations or types of measurements or remote sensing data within a given network seeks to identify the value ranking of information sources, accounting for analysis achieved. This problem has been investigated by Cardinali et al. (2004), Cardinali (2009), and a sequence of related papers , or Liu and Kalnay (2008), the latter without use of an adjoint model. A related approach was described by Baker and Daley (2000), who exploited sensitivities to

10  observations to identify their spatial extensions of impact.

Finally, the need to quantify the information content provided by the observations (iii) can be satisfied by suitable and calculable measures, such as entropy reduction or DFS. The concept of DFS has been applied to satellite retrieval problems, typically of lower dimensions as compared to data assimilation (see for example Eyre 1990; Rodgers 2000; Rabier et al. 2002; Fourrié et al. 2003; Martynenko et al. 2010; Fisher 2003 ). However, these studies focused on the classical data assimilation

15  problem with initial values or prognostic state variables as the only parameters to be optimized. Yet for chemistry transport or greenhouse gas models, which highly depend on the emissions in the troposphere, the optimization of the initial state is no longer the only issue. Rather, the optimization of emissions play an equally important role as initial values. In order to get better analysis from combining the model with observations, efforts of joint optimization by adding the emission rates to concentrations have been made (Elbern et al. 2000, 2007; Bocquet 2012; Bocquet and Sakov 2013; Miyazaki et al. 2012;

20  Tang et al. 2011, 2013; Winiarek et al. 2014). Yet the lack of ability to observe and estimate surface emission fluxes directly is a major roadblock, hampering the progress in predictive skills of climate and atmospheric chemistry models. Therefore, the capacity to distinguish between the degree of freedom for signal of both emission rates and concentrations is crucial to assess the value of a measurement network. These assessment results are dependent not only on observation network and its deployment with respect to emission sources, but also influenced by the assimilation window lengths and the meteorological

25  transport conditions.

In this context a meanwhile classical task is greenhouse gas inversion, aiming at the estimation of carbon dioxide, methane, and nitrous oxide sources, from which a rich set of literature emerged (e.g Peter et al. 2005). The sensitivity of the model evolution with respect to the model errors and the observation network as its detector is a key quantity to be analysed. Several methodologies have been proposed to account for model errors in both variational and ensemble data assimilation (e.g. Bellsky

30  et al. 2014; Gillijns and De Moor 2007; Li et al. 2009; Smith et al. 2013; Tremolet 2007; Daescu 2004; Zupanski 2007). Navon (1997) outlined the perceptibility and stability in optimal parameter estimation in meteorology and oceanography. Elbern et al. (2000, 2007) took the strong constraint by a given diurnal profile shape of emission rates such that their amplitudes in addition to initial values are the only parameters to be optimized by 4D-variational inversion. A general framework to optimize a set of parameters controlling the 4D-var data assimilation system was introduced by Cioaca and Sandu (2014) and applied to shallow

35  model state and other parameters in a related paper (Cioaca and Sandu 2014).

Singular value decomposition (SVD) is a well-known tool applied to identifying the priorities of observations by detecting the fastest growing uncertainties in meteorological models (e.g Lorenz 1965; Buizza and Palmer 1995; Khattatov et al. 1999; Johnson 2003; Liao et al. 2006; Daescu 2008; Abida and Bocquet 2009; Kang and Xu 2012; Singh et al. 2012, 2013; Sandu et al. 2013). 
[revised manuscript text omitted]

20  Appendix A, a more general case of the transport model extended by emission is shown.

**3   The degree of freedom for signal of concentrations and emissions**

In this section we will introduce the theoretical approach to determine the DFS of concentrations and emissions, resting on the extended model in Section 2. This approach gives us access to determine the efficiency of observations to optimize each variable based on the Kalman smoother within a finite-time interval.

25  For convenience, we generalize the atmospheric transport model (10) by the following discrete-time linear system on the time interval $[t_0, t_1, \cdots, t_N]$:

$$x(t_{k+1}) = M(t_{k+1}, t_k)x(t_k) + \varepsilon(t_k), \tag{12}$$

$$y(t_k) = H(t_k)x(t_k) + \nu(t_k), \tag{13}$$

where $x(\cdot) \in \mathbf{R}^n$ is the state variable and $y(t_k) \in \mathbf{R}^{m(t_k)}$ is the observation vector at time $t_k$. The model error $\varepsilon(t_k)$ and the observation error $\nu(t_k)$, $k = 1, \cdots, N$ of Gaussian distributions have zero means. The model error covariance matrix is denoted by $Q(t_k)$, while the observation error covariance matrix is denoted by $R(t_k)$.

5  We denote the BLUE of $x(t_i)$ based on $\{y(t_0), \cdots, y(t_k)\}$ by $\hat{x}(t_i|t_k)$, $t_i, t_k \in [t_0, \cdots, t_N]$. Especially, the prior estimation of $x(t_0)$ is denoted by $\hat{x}(t_0|t_{-1})$. Correspondingly, $P(t_i|t_k)$ is defined as the error covariance of $\hat{x}(t_i|t_k)$, $t_i \in [t_0, \cdots, t_N]$ and $t_k \in [t_{-1}, \cdots, t_N]$. It is known that the inverse of the analysis error covariance matrix at initial time, $P^{-1}(t_0|t_N)$ of a fixed-interval Kalman smoother is the optimal Hessian of the underlying cost function of 4D-Var (Li and Navon, 2001). Thus, we have

$$\quad P^{-1}(t_0|t_N) = P^{-1}(t_0|t_{-1}) + \sum_{i=0}^{N} M^{\top}(t_i, t_0) H^{\top}(t_i) R^{-1}(t_i) H(t_i) M(t_i, t_0). \tag{14}$$

It is clear that (14) comprises the information of the initial condition, model evolution, observation configurations and errors over the entire time interval $[t_0, \cdots, t_N]$. At the same time, it is independent of any specific data and state vector, apart from the reference model evolution $M(\cdot, \cdot)$ needed for the linearization, as well as the observation operator $H(\cdot)$. Actually, if we define

$$\mathcal{G} = \begin{pmatrix} H(t_0)M(t_0, t_0) \\ H(t_1)M(t_1, t_0) \\ \vdots \\ H(t_N)M(t_N, t_0) \end{pmatrix}, \quad \mathcal{R}^{-1} = \begin{pmatrix} R^{-1}(t_0) & & & \\ & R^{-1}(t_1) & & \\ & & \ddots & \\ & & & R^{-1}(t_N) \end{pmatrix}, \tag{15}$$

15 we can rewrite (14) as

$$P^{-1}(t_0|t_N) = P^{-1}(t_0|t_{-1}) + \mathcal{G}^{\top}\mathcal{R}^{-1}\mathcal{G}, \tag{16}$$

where $\mathcal{G}^{\top}R^{-1}\mathcal{G}$ is the observability Gramian with respect to $\mathcal{R}^{-1}$ in control theory (Brockett, 1994). It represents the observation capacity of the observation networks with respect to the model.

Though (16) meets the demand to represent the estimate covariance by all available information before starting the data
20 assimilation procedure, it cannot be applied directly to evaluate the potential improvement of the estimate by the Kalman smoother, due to the lack of clear statistical significance of the inverse of a covariance matrix. We aspire a matrix, which allows us for a direct and normalized comparison between sensitivities to initial values and emission rates. To this end, we consider matrix $\tilde{P}$ with the following form:

$$\tilde{P} = P^{-\frac{1}{2}}(t_0|t_{-1})(P(t_0|t_{-1}) - P(t_0|t_N))P^{-\frac{1}{2}}(t_0|t_{-1}) = I - P^{-\frac{1}{2}}(t_0|t_{-1})P(t_0|t_N)P^{-\frac{1}{2}}(t_0|t_{-1}), \tag{17}$$

25 where $I$ is the identity matrix and $P^{\frac{1}{2}}(t_0|t_{-1})$ satisfies $P^{\frac{1}{2}}(t_0|t_{-1})P^{\frac{1}{2}}(t_0|t_{-1}) = P(t_0|t_{-1})$.

The matrix $\tilde{P}$ is a normalized matrix of the difference between the background forecast error covariance matrix $P(t_0|t_{-1})$ and the analysis error covariance matrix $P(t_0|t_N)$, as inferred the Kalman smoother. It shows how much the observation

networks improve the estimation of model states and is the foundation matrix to study the DFS of models(Fisher 2003; Rodgers 2000; Singh et al. 2013).

Since $P(t_0|t_N)$ is unknown prior to the data assimilation procedure, we use (16) to rewrite $\tilde{P}$ as

$$
\begin{aligned}
\tilde{P} &= P^{-\frac{1}{2}}(t_0|t_{-1})(P(t_0|t_{-1}) - P(t_0|t_N))P^{-\frac{1}{2}}(t_0|t_{-1}) \\
&= P^{-\frac{1}{2}}(t_0|t_{-1})(P(t_0|t_{-1}) - (P^{-1}(t_0|t_{-1}) + \mathcal{G}^\top \mathcal{R}^{-1}\mathcal{G})^{-1})P^{-\frac{1}{2}}(t_0|t_{-1}) \\
&= I - P^{-\frac{1}{2}}(t_0|t_{-1})(P^{-1}(t_0|t_{-1}) + \mathcal{G}^\top \mathcal{R}^{-1}\mathcal{G})^{-1}P^{-\frac{1}{2}}(t_0|t_{-1}) \\
&= I - (I + P^{\frac{1}{2}}(t_0|t_{-1})\mathcal{G}^\top \mathcal{R}^{-1}\mathcal{G}P^{\frac{1}{2}}(t_0|t_{-1}))^{-1}.
\end{aligned}
\tag{18}
$$

It is worth noting that in (18)

$$
I + P^{\frac{1}{2}}(t_0|t_{-1})\mathcal{G}^\top \mathcal{R}^{-1}\mathcal{G}P^{\frac{1}{2}}(t_0|t_{-1})
\tag{19}
$$

is always invertible even if the observation Gramian $\mathcal{G}^\top \mathcal{G}$ is not full-rank. Thus, $\tilde{P}$ is well-defined for all models with invertible initial covariance and observation systems with invertible error covariances within assimilation window $t_0$ to $t_N$. Then, we apply the singular value decomposition to simplify (18)

$$
P^{\frac{1}{2}}(t_0|t_{-1})\mathcal{G}^\top \mathcal{R}^{-\frac{1}{2}} = VSU^\top,
\tag{20}
$$

where $V$ and $U$ are unitary matrices consisting of the left and right singular vectors,respectively, while $S$ is the rectangular diagonal matrix consisting of the singular values.

Then, (18) can be simplified as

$$
\begin{aligned}
\tilde{P} &= I - (I + P^{\frac{1}{2}}(t_0|t_{-1})\mathcal{G}^\top \mathcal{R}^{-1}\mathcal{G}P^{\frac{1}{2}}(t_0|t_{-1}))^{-1} \\
&= I - (I + VSS^\top V^\top)^{-1} \\
&= VV^\top - (VV^\top + VSS^\top V^\top)^{-1} \\
&= VV^\top - (V(I + SS^\top)V^\top)^{-1} \\
&= V(I - (I + SS^\top)^{-1})V^\top \\
&= \sum_{i=1}^{r} \frac{s_i^2}{1 + s_i^2} v_i v_i^\top,
\end{aligned}
\tag{21}
$$

where $r$ is the rank of (18) and $v_i$ is the $i^{th}$ left singular vector in $V$ related to the singular value $s_i$, which is the $i^{th}$ element on the diagonal of $S$.

It is clear that the sum of the diagonal entries of $\tilde{P}$ can be used to evaluate the total improvements of model states. Thus, the nuclear norm is appropriately taken as the metric, which is defined as

$$
\|A\|_1 = \mathrm{tr}(\sqrt{A^\top A}),
\tag{22}
$$

where $A$ is any matrix and $\mathrm{tr}(\cdot)$ denotes the trace of the matrix.

From (21), we obtain

$$\|\tilde{P}\|_1 = \text{tr}(\tilde{P}) = \sum_{i=1}^{r} \frac{s_i^2}{1+s_i^2}. \tag{23}$$

This is well-known as the degree of freedom for signal (DFS) of the model (e.g Rodgers 2000).

It is obvious that $\|\tilde{P}\|_1 < \|I\|_1 = n$. Here $n$ can be considered as the total relative 
[revised manuscript text omitted]

where $\bar{U} \in \mathbf{R}^{m \times m}$ and $\bar{V} \in \mathbf{R}^{n \times n}$ consists of the eigenvectors of $\mathcal{R}^{-\frac{1}{2}} \mathcal{G} \bar{P}(t_0|t_{-1}) \mathcal{G}^\top \mathcal{R}^{-\frac{1}{2}}$ and $\bar{P}^{\frac{1}{2}}(t_0|t_{-1}) \mathcal{G}^\top \mathcal{R}^{-1} \mathcal{G} \bar{P}^{\frac{1}{2}}(t_0|t_{-1})$, respectively. $S \in \mathbf{R}^{n \times m}$ consists of the singular values on its diagonal.

We denote the rank of (50) by $r$. Then, we rewrite $\bar{P}$ as

$$
\begin{aligned}
\bar{P} &= \bar{P}^{\dagger \frac{1}{2}}(t_0|t_{-1})(\bar{P}(t_0|t_{-1}) - \bar{P}(t_0|t_N))\bar{P}^{\dagger \frac{1}{2}}(t_0|t_{-1}) \\
&= \bar{V}\bar{S}^\top \bar{U}^\top(\bar{U}\bar{U}^\top + \bar{U}(\bar{S}\bar{S}^\top)\bar{U}^\top)^{-1}\bar{U}\bar{S}\bar{V}^\top \\
&= \bar{V}\bar{S}^\top(I + \bar{S}^\top \bar{S})^{-1}\bar{S}\bar{V}^\top \\
&= \sum_{i=1}^{r} \frac{\bar{s}_i^2}{1 + \bar{s}_i^2}\bar{v}_i \bar{v}_i^\top.
\end{aligned}
\tag{51}
$$

where $r$ is the rank of $\bar{P}$ and $\bar{v}_i$ is the $i^{th}$ left singular vector in $V$ related to the singular value $\bar{s}_i$, which is the $i^{th}$ element on the diagonal of $S$.

We observe that (51) and (21) have a similar form. By virtue of

$$
\bar{P}^{\dagger \frac{1}{2}}(t_0|t_{-1})\bar{P}_{xy}^f \bar{\mathcal{R}}^{-\frac{1}{2}} = \bar{P}^{\frac{1}{2}}(t_0|t_{-1})\mathcal{G}^\top \mathcal{R}^{-\frac{1}{2}},
\tag{52}
$$

the final results of (21) and (51) are equivalent. However, compared with $P^{\frac{1}{2}}(t_0|t_{-1})\mathcal{G}^\top \mathcal{R}^{-\frac{1}{2}}$, the ensemble expression $\bar{P}^{\dagger \frac{1}{2}}(t_0|t_{-1})\bar{P}_{xy}^f \bar{\mathcal{R}}^{-\frac{1}{2}}$ processes the absolute advantage that in the calculation of $\bar{P}_{xy}^f$ since we do not need the explicit form of $\mathcal{G}$. It allows us to code it line by line such that our approach is computationally more efficient.

Analog to the standard case, we can similarly define *the ensemble degree of freedom for signal* (EnDFS) as $\|\bar{P}\|_1$ and consider each element on the diagonal of $\bar{P}$ as *the contribution to EnDFS* of the corresponding model state.

Since $\bar{P}(t_0|t_{-1})$ is typically not full rank,

$$
\begin{aligned}
&\bar{P}^{\dagger \frac{1}{2}}(t_0|t_{-1})(\bar{P}(t_0|t_{-1}) - \bar{P}(t_0|t_N))\bar{P}^{\dagger \frac{1}{2}}(t_0|t_{-1}) \\
&= \bar{P}^{\dagger \frac{1}{2}}(t_0|t_{-1})\bar{P}(t_0|t_{-1})\bar{P}^{\dagger \frac{1}{2}}(t_0|t_{-1}) - \bar{P}^{\dagger \frac{1}{2}}(t_0|t_{-1})\bar{P}(t_0|t_N)\bar{P}^{\dagger \frac{1}{2}}(t_0|t_{-1}) \\
&= V_0 \hat{S}_0^\dagger V_0^\top(V_0 \hat{S}_0^2 V_0^\top)V_0 \hat{S}_0^\dagger V_0^\top - \bar{P}^{\dagger \frac{1}{2}}(t_0|t_{-1})\bar{P}(t_0|t_N)\bar{P}^{\dagger \frac{1}{2}}(t_0|t_{-1}) \\
&= V_0 I_{r_0} V_0^\top - \bar{P}^{\dagger \frac{1}{2}}(t_0|t_{-1})\bar{P}(t_0|t_N)\bar{P}^{\dagger \frac{1}{2}}(t_0|t_{-1}),
\end{aligned}
\tag{53}
$$

where $I_{r_0}$ is the diagonal matrix with the diagonal $(\mathbf{1}_{1 \times r_0}, 0_{1 \times (n-r_0)})$. It is clear from (48) that $\bar{P}^{\dagger \frac{1}{2}}(t_0|t_{-1})\bar{P}(t_0|t_N)\bar{P}^{\dagger \frac{1}{2}}(t_0|t_{-1})$ is still a nonnegative definite matrix.

Thus, the *ensemble relative degree of freedom for signal*(EnRDFS) is defined by

$$
\bar{p} = \frac{\|\bar{P}\|_1}{\|I_{r_0}\|_1} = \frac{\|\bar{P}\|_1}{r_0} \in [0,1).
\tag{54}
$$

In order to distinguish the improvements for concentrations and emission rates, the *ensemble relative ratios of DFS* remain

$$
\bar{p}^c = \frac{\|\bar{P}^c\|_1}{\|\bar{P}\|_1}, \quad \bar{p}^e = \frac{\|\bar{P}^e\|_1}{\|\bar{P}\|_1}.
\tag{55}
$$

If we further consider the nonlinear dynamic model, we can renew the definition of the forecasting observation configurations as

$$
y_k^f = \mathcal{G}(\hat{x}_k(t_0|t_{-1})), \quad k = 1, \cdots, q,
\tag{56}
$$

such that it can follow the nonlinear model, where $\mathcal{G}$ is again a combined model-observation nonlinear operator.

Correspondingly, the ensemble mean of $\bar{y}_k^f$ and $\bar{P}_{xy}^f$ can be calculated based on (56) with the nonlinear $\mathcal{G}$. Thus, the above ensemble-based approach is available for nonlinear models.

**5 Sensitivity of observation networks**

5    The above discussion about DFS aims to evaluate a predefined measurement network on its potential to analyze initial values and emission rates simultaneously. In Appendix B, independent of any concrete data assimilation method, we use the singular vector approach to identify the sensitive directions of observation networks to initial values and emission rates and show the association between the efficiency and sensitivity of observation networks.

From Appendix B, we can see that the singular value $s_k$ shows the amplification of the impact of the initial state to the

10    observation configurations during the entire time interval. The associated singular vector in the state space $v_k$ is the direction of $k^{\text{th}}$ growth of the perturbation of observations evolved from the initial perturbation. With the special choice $W_0 = P^{-1}(t_0|t_{-1})$ and $\mathcal{W} = \mathcal{R}^{-1}$ we compare the sensitivity analysis with the analysis in Section 3. Since $\frac{s_k^2}{1+s_k^2}$ are decreasing with the decrease of $s_k$, $k = 1, \cdots, n$, it is clear that $v_k$ is also the $k^{\text{th}}$ direction which maximizes the relative improvement of estimates based on Kalman smoother. It indicates that the states contributing to DFS more are the same with the states more sensitive to the

15    observation networks. Besides, the leading singular value $s_1$ is related to the operator norm of $\tilde{P}$ as

$$\|\tilde{P}\| = \max_{\|x\|=1} \|\tilde{P}x\| = \frac{s_1^2}{1+s_1^2}, \tag{57}$$

which implies the upper boundedness of $\tilde{P}$. It gives us an access to approximate and target the sensitive parameters or areas with the metric of the leading singular vectors weighted by the corresponding singular values.

Moreover, due to the homogeneity of the atmospheric transport model state vector extended with emissions, the above

20    sensitivity analysis can be easily applied by dividing singular vectors into the block form according to the dimensions of the initial state and emissions. The corresponding block parts of different singular vectors indicate the different sensitive directions of the initial state and emissions and allow for this relative quantification. Correspondingly, we can approximate and target the parameters sensitive to the existing observation networks for both initial values and emission rates.

**6 Example**

25    We consider a linear advection-diffusion model with Dirichlet horizontal boundary condition and Neumann boundary condition in the vertical direction on the domain $[0, 14] \times [0, 14] \times [0, 4]$,

$$\frac{\partial \delta c}{\partial t} = -v_x \frac{\partial \delta c}{\partial x} - v_y \frac{\partial \delta c}{\partial y} + \frac{\partial}{\partial z}\left(K(z)\frac{\partial \delta c}{\partial z}\right) + \delta e, \tag{58}$$

where $\delta c$, $\delta e$ are the perturbations of the concentration, the emission rate of a species respectively. For vertical diffusion, $K(z)$ is a differentiable function of height $z$.

For velocity $v_x = v_y = 0.5$ and the time step $\triangle t = 0.5$, the numerical solution is based on the symmetric operator splitting technique (Yanenko, 1971) with the following operator sequence

$$\delta c(t + \triangle t) = T_x T_y D_z A D_z T_y T_x \delta c(t), \tag{59}$$

where $T_x$ and $T_y$ are transport operators in horizontal directions $x$ and $y$, $D_z$ is the diffusion operator in vertical direction $z$. The parameters of emission and deposition rates are included in $A$. The Lax-Wendroff algorithm is chosen as the discretization method for horizontal advection with $\triangle x = \triangle y = 1$. The vertical diffusion is discretized with $\triangle z = 1$ by Crank-Nicolson scheme with the Thomas algorithm (Higham, 2002) as solver. The number of the grid points is $N_g = 1125$.

With the same temporal and spacial discretization of the concentration, the background knowledge of the emission rate is given by $e_b(t_n, i, j, l)$, where $n = 1, \cdots, N$. We establish the discrete dynamic model of the emission rate according to (8)

$$\delta e(t_{n+1}) = M_e(t_{n+1}, t_n)\delta e(t_n), \quad n = 1, \cdots, N, \tag{60}$$

where $M_e(t_{n+1}, t_n) = e_b(t_{n+1})/e_b(t_n)$.

For expository reasons we assume $\delta d$ is a constant over time and the only one fixed observation configuration is time-invariant. It indicates that the observation operator mapping the state space to the observation space is a $1 \times 2N_g$ time-invariant matrix.

In our simulations, we produce $q = 500$ (the ensemble number) samples for the initial concentration and emission rate respectively by pseudo independent random numbers and make the states correlated by the moving average technique. It has been tested that the computation cost of our approach is linearly increasing with the number of ensembles. In the following, we present three different tests, aiming to demonstrate the roles of variable winds, emissions, and vertical diffusion.

*Advection test:* The following part demonstrates the application of the DFS analysis tool by basic examples, designed to show the expected elementary outcomes of the following situations, which exhibit the effects of assimilation window length in relation to emission location: these include (i) an assimilation window is too short to capture emission impacts at the observation site, (ii) an extended assimilation window with balanced signal of impacts of concentrations and emissions at the observation site, and (iii) a further increased assimilation window features a declining impact of initial values and growing emission impact. The first elementary advection test (Fig. 1 to Fig. 7) identifies the sensitivities of parameters subject to different wind direction and data assimilation window (DAW) through the DFS. Focusing on the advection effects, we apply the model with a weak diffusion process $(K(z) = 0.5e^{-z^2})$.

In Fig. 1 to 3 we assume southwesterly winds and the assigned data assimilation windows are $10\triangle t$, $35\triangle t$ and $48\triangle t$ respectively. The computation times are approximately 8.1s, 28.5s and 39.4s in our tests with the above three different assimilation windows, from which we can verify that the computation cost is nearly linearly increasing with the length of data assimilation window. 
[revised manuscript text omitted]

**7    Conclusions and Outlooks**

This study demonstrates the quantification of the sensitivity of a given measurement network to impact initial trace gas state and emission rates for transport-diffusion models forced by emission. The indicators adopt the degrees of freedom for signal concept. Resting on a Kalman smoother, the contribution to the degree of freedom for signal is derived as the criterion to evaluate the potential improvement of the extended state vector as calculated by singular value decomposition. With a statistical interpretation, we can apply it to determine in advance, which parameters can be optimized by the data assimilation procedure. The degree of freedom for signal and a number of metrics provide us with the quantitative solutions to measure to what extend the parameters can be optimized. Due to its normalization, it is uniformly available for any prior initial values of invertible background covariances. Further, the proposal of the ensemble based relative improvement covariance, based on EnKS, gives us a computationally feasible access to assess the degree of freedom for signal.

The sensitivity of observational networks was formulated by seeking the fastest directions of the perturbation ratio between initial states and observation configurations during the entire time interval. An elementary advection-diffusion example illustrated the significance of relative improvements covariances and their various metrics in different situations and compared them with the results of the sensitivity analysis.

In the future, it is planed to apply the efficiency analysis into the real atmospheric transport model to solve practical problems, also for nonlinear reactive chemistry-transport-diffusion models, as far as the validity of the tangent linear assumption holds, exactly as in atmospheric chemistry data assimilation problems. It is expected that we will get deeper insight to the sensitivity analysis for wider applications. For example, in order to evaluate the impact of observations in some certain locations, the local projection operator introduced by Buizza and Montani (1999) can be applied into approaches presented in Section 4 and Section 5.

**8 Code availability**

The codes are available via "https://www.geosci-model-dev-discuss.net/gmd-2017-220/gmd-2017-220-supplement.zip" , or from the corresponding author on request.

**Appendix A**

In this appendix, we show a more general case to extend the control vector by emissions. As we know, the initial state and emission rates do not have the same dimension in some practical cases, as there can be more than one different kinds of emissions for one species. Compared with (2), the general situation leads us to consider the following model

$$\frac{d\delta c}{dt} = \mathbf{A}\delta c + B(t)\delta e(t),\tag{61}$$

where $B(t)$ is an operator transforming the emission state vector into the concentration-state space. Combining with (8), we obtain the extended model

$$\begin{pmatrix} \delta c(t) \\ \delta e(t) \end{pmatrix} = \begin{pmatrix} M(t,t_0) & \int_{t_0}^{t} M(t,s)B(s)M_e(s,t_0)ds \\ 0 & M_e(t,t_0) \end{pmatrix} \begin{pmatrix} \delta c(t_0) \\ \delta e(t_0) \end{pmatrix}.\tag{62}$$

**Appendix B**

In this appendix, we will introduce the singular vector approach to identify the sensitive directions of initial values and emission rates to the observation netvorks and show the association between the efficiency and sensitivity of observation networks.

We define the magnitude of the perturbation of the initial state by the norm in the state space with respect to a positive definite matrix $W_0$, typically the forecast error covariance matrix.

$$\|\delta x(t_0)\|_{W_0}^2 = \langle \delta x(t_0), W_0 \delta x(t_0)\rangle.\tag{63}$$

Likewise, we define the magnitude of the related observations perturbation in the time interval $[t_0, \cdots, t_N]$ by the norm with respect to a sequence of positive definite matrices $\{W(t_k)\}_{k=1}^{N}$

$$\|\delta y_c\|_{\{W(t_k)\}}^2 = \sum_{k=0}^{N} \langle \delta y_c(t_k), W(t_k)\delta y_c(t_k)\rangle,\tag{64}$$

where $\delta y_c = (\delta y_c^\top(t_0), \cdots, \delta y_c^\top(t_N))^\top$.

In order to find the direction of observation configuration which minimizes the perturbation of the initial states, the ratio

$$\frac{\|\delta x(t_0)\|_{W_0}^2}{\|\delta y_c\|_{\{W(t_k)\}}^2}, \quad \delta y \neq 0_{m\times 1}.\tag{65}$$

must be minimized. It is equivalent to maximize the ratio of the magnitude of observation perturbation and the initial perturbation

$$\frac{\|\delta y_c\|_{\{W(t_k)\}}^2}{\|\delta x(t_0)\|_{W_0}^2}, \quad \delta x(t_0) \neq 0_{n\times 1}.\tag{66}$$

Thus, we define the measure the perturbation growth as

$$g^2 = \frac{\|\delta y_c\|^2_{\{W(t_k)\}}}{\|\delta x(t_0)\|^2_{W_0}} \tag{67}$$

$$= \sum_{k=0}^{N} \frac{\langle \delta y_c(t_k), W(t_k)\delta y_c(t_k)\rangle}{\langle \delta x(t_0), W_0\delta x(t_0)\rangle}$$

$$= \sum_{k=0}^{N} \frac{\langle H(t_k)\delta x(t_k), W(t_k)H(t_k)\delta x(t_k)\rangle}{\langle \delta x(t_0), W_0\delta x(t_0)\rangle}$$

$$= \sum_{k=0}^{N} \frac{\langle \delta x(t_k), H(t_k)^\top W(t_k)H(t_k)\delta x(t_k)\rangle}{\langle \delta x(t_0), W_0\delta x(t_0)\rangle}$$

$$= \sum_{k=0}^{N} \frac{\langle \delta x(t_0), M(t_k,t_0)^\top H(t_k)W(t_k)H(t_k)M(t_k,t_0)\delta x(t_0)\rangle}{\langle \delta x(t_0), W_0\delta x(t_0)\rangle}$$

$$= \frac{\langle \delta x(t_0), \sum_{k=0}^{N} M(t_k,t_0)^\top H(t_k)^\top W(t_k)H(t_k)M(t_k,t_0)\delta x(t_0)\rangle}{\langle \delta x(t_0), W_0\delta x(t_0)\rangle}$$

$$= \frac{\langle \delta x(t_0), \mathcal{G}^\top \mathcal{W}\mathcal{G}\delta x(t_0)\rangle}{\langle \delta x(t_0), W_0\delta x(t_0)\rangle}, \quad \delta x(t_0) \neq 0, \tag{68}$$

where $\mathcal{G}$ has the same definition as given in Section 3, and

$$\mathcal{W} = \begin{pmatrix} W(t_0) & & \\ & \ddots & \\ & & W(t_N) \end{pmatrix}. \tag{69}$$

We arrive at the eigenvalue problems (Liao et al., 2006)

$$W_0^{-\frac{1}{2}}\mathcal{G}^\top \mathcal{W}\mathcal{G}W_0^{-\frac{1}{2}}v_k = s_k^2 v_k, \quad W^{\frac{1}{2}}\mathcal{G}W_0^{-1}\mathcal{G}^\top W^{\frac{1}{2}}u_k = s_k^2 u_k, \tag{70}$$

where $s_1 \geqslant s_2 \geqslant \cdots \geqslant s_n \geqslant 0$ are singular values, $\{v_k\}_{k=1}^n$ and $\{u_k\}_{k=1}^n$ are the corresponding orthogonal singular vectors. Then, $\max_{\delta x(t_0)\neq 0} g^2 = s_1^2$.

Especially, if the perturbation norms are provided by the choice $W_0 = P^{-1}(t_0|t_{-1})$ and $\mathcal{W} = \mathcal{R}^{-1}$,

$$g^2 = \frac{\langle \delta x(t_0), \mathcal{G}^\top \mathcal{R}^{-1}\mathcal{G}\delta x(t_0)\rangle}{\langle \delta x(t_0), P^{-1}(t_0|t_{-1})\delta x(t_0)\rangle}, \quad \delta x(t_0) \neq 0. \tag{71}$$

Correspondingly, we consider the singular value decomposition of the following matrices

$$P^{\frac{1}{2}}(t_0|t_{-1})\mathcal{G}^\top \mathcal{R}^{-1}\mathcal{G}P^{\frac{1}{2}}(t_0|t_{-1})v_k = s_k^2 v_k; \tag{72}$$

$$\mathcal{R}^{-\frac{1}{2}}\mathcal{G}P(t_0|t_{-1})\mathcal{G}^\top \mathcal{R}^{-\frac{1}{2}}u_k = s_k^2 u_k, \quad k = 1, \cdots, n, \tag{73}$$

where $s_k$, $v_k$ and $u_k$, $k = 1,\ldots,n$ denotes the singular values and corresponding singular vectors to simplify the notation. At the same time, it can be seen that the above singular value decomposition problem coincides with (20) in Section 3.

*Acknowledgements.* This study was supported by HITEC (Helmholtz Interdisciplinary Doctoral Training in Energy and Climate Research). HITEC is the Graduate School of Forschungszentrum Jülich and the five partner universities Aachen, Bochum, Cologne, Düsseldorf and Wuppertal focusing on energy and climate research. Technical and computational support was provided by the Jülich Supercomputer Centre (JSC) of the Research Centre Jülich. A large and central part of the case studies has been computed on JSC supercomputer JUECA. Access given to these computational resources is highly appreciated.

[revised manuscript text omitted]

---

## Author Comment (AC5) · 30 Aug 2018

The comment was uploaded in the form of a supplement:
https://www.geosci-model-dev-discuss.net/gmd-2017-220/gmd-2017-220-AC5-supplement.pdf